# Zebrafish behavioural profiling identifies GABA and serotonin receptor ligands related to sedation and paradoxical excitation

Matthew N. McCarroll[1,11], Leo Gendelev[1,11], Reid Kinser[1], Jack Taylor [1], Giancarlo Bruni [2,3], Douglas Myers-Turnbull [1], Cole Helsell[1], Amanda Carbajal[4], Capria Rinaldi[1], Hye Jin Kang[5], Jung Ho Gong[6], Jason K. Sello[6], Susumu Tomita[7], Randall T. Peterson[2,10], Michael J. Keiser [1,8] & David Kokel[1,9]

Anesthetics are generally associated with sedation, but some anesthetics can also increase brain and motor activity—a phenomenon known as paradoxical excitation. Previous studies have identified $GABA_A$ receptors as the primary targets of most anesthetic drugs, but how these compounds produce paradoxical excitation is poorly understood. To identify and understand such compounds, we applied a behavior-based drug profiling approach. Here, we show that a subset of central nervous system depressants cause paradoxical excitation in zebrafish. Using this behavior as a readout, we screened thousands of compounds and identified dozens of hits that caused paradoxical excitation. Many hit compounds modulated human $GABA_A$ receptors, while others appeared to modulate different neuronal targets, including the human serotonin-6 receptor. Ligands at these receptors generally decreased neuronal activity, but paradoxically increased activity in the caudal hindbrain. Together, these studies identify ligands, targets, and neurons affecting sedation and paradoxical excitation in vivo in zebrafish.

[1] Institute for Neurodegenerative Diseases, University of California, San Francisco, CA 94143, USA. [2] Cardiovascular Research Center and Division of Cardiology, Department of Medicine, Massachusetts General Hospital, Harvard Medical School, Charlestown, MA 02129, USA. [3] Department of Systems Biology, Harvard Medical School, 149 13th Street, Charlestown, MA 02129, USA. [4] Department of Biology, San Francisco State University, San Francisco, CA, USA. [5] Department of Pharmacology and NIMH Psychoactive Drug Screening Program, University of North Carolina Chapel Hill Medical School, Chapel Hill, NC 27759, USA. [6] Department of Chemistry, Brown University, Providence, RI 02912, USA. [7] Department of Cellular and Molecular Physiology, Department of Neuroscience, Yale University School of Medicine, New Haven, CT 06510, USA. [8] Departments of Pharmaceutical Chemistry and of Bioengineering & Therapeutic Sciences, University of California, San Francisco, CA 94158, USA. [9] Department of Physiology, University of California, San Francisco, CA 94158, USA. [10] Present address: Department of Pharmacology and Toxicology, College of Pharmacy, University of Utah, Salt Lake City, UT 84112, USA. [11] These authors contributed equally: Matthew N. McCarroll, Leo Gendelev. Correspondence and requests for materials should be addressed to M.J.K. (email: keiser@keiserlab.org) or to D.K. (email: david.kokel@ucsf.edu)

Anesthetics and other central nervous system (CNS) depressants primarily suppress neural activity, but sometimes they also cause paradoxical excitation[1]. During paradoxical excitation, brain activity increases[2,3] and produces clinical features such as confusion, anxiety, aggression, suicidal behavior, seizures, and aggravated rage[4,5]. These symptoms primarily affect small but vulnerable patient populations including psychiatric, pediatric, and elderly patients[6,7]. Understanding paradoxical excitation is important for discovering, understanding, and developing CNS depressants and for understanding how small molecules affect the vertebrate nervous system. However, relatively few compounds have been identified that cause paradoxical excitation, and few model systems have been identified that reproducibly model paradoxical excitation in vivo.

Many ligands that cause paradoxical excitation are agonists or positive allosteric modulators (PAMs) of GABA$_A$ receptors (GABA$_A$Rs), the major type of inhibitory receptors in the CNS[8]. However, it is likely that other mechanisms also affect paradoxical excitation. One such mechanism may involve serotonin imbalance, which affects behavioral disinhibition[9,10], and has paradoxical effects on neuronal circuit output[11]. For example, the serotonin-6 receptor (HTR6) is an excitatory G protein-coupled receptor (GPCR) reported to modulate cholinergic and glutamatergic systems by disinhibiting GABAergic neurons[12]. In serotonin syndrome, excessive serotonergic signaling causes agitation, convulsions, and muscle rigidity. Despite these excitatory effects of excessive serotonergic signaling, several serotonin receptor agonists are used as anxiolytics, hypnotics, and anticonvulsants[13]. Examples include clemizole and fenfluramine, which promote 5-HT signaling and have anticonvulsant properties in humans and zebrafish[13,14]. By contrast, serotonin antagonists and inverse agonists improve sleep and are used for treating insomnia[15]. Furthermore, serotonin receptors are secondary and tertiary targets of some anesthetics, suggesting that 5-HT receptors may contribute to sedation[16]. However, the potential impact of serotonin receptors on anesthesia and paradoxical excitation is poorly understood.

In principle, large-scale behavior-based chemical screens would be a powerful way to identify compounds that cause sedation and paradoxical excitation. The reason is that phenotype-based screens are not restricted to predefined target-based assays. Rather, phenotype-based screens can be used to identify targets and pathways that produce poorly understood phenotypes. Indeed, virtually all CNS and anesthetic drug prototypes were originally discovered based on their behavioral effects before their targets were known[17]. Furthermore, these compounds were valuable tools for understanding the mechanisms of anesthesia and sedation. Although behavior-based chemical screens in vertebrates would be most relevant for human biology, behavioral assays in mice, primates, and other mammals are difficult to scale.

Zebrafish are uniquely well-suited for studying the chemical biology of sedation and paradoxical excitation. Zebrafish are vertebrate animals with complex brains and behaviors, they are small enough to fit in 96-well plates, and they readily absorb compounds dissolved in the fish water. Compared with humans, zebrafish share many conserved genes and neurotransmitter signaling pathways[18]. For example, the zebrafish genome contains orthologs for all but two human GABA$_A$R subunit isoforms[19]. The α-isoform family is the largest and most diverse family of GABA$_A$R subunits in both humans and zebrafish[20]. The zebrafish genome also encodes orthologs of serotonin receptors, including orthologs of HTR6[21,22]. Additionally, there are several important differences between the species. One such difference is that the zebrafish genome encodes a GABA$_A$R β$_4$-subunit, which does not have a clear ortholog in mammals[19]. Another difference is that whereas mammals have six GABA$_A$R α-subunit isoforms,

zebrafish have eight[20]. Previously, drug profiling studies in zebrafish have identified neuroactive compounds related to antipsychotics, fear, sleep, and learning[23–26]. However, specific behavioral profiles for compounds that cause paradoxical excitation have not been previously described in zebrafish.

The purpose of these studies is to identify and understand compounds that cause paradoxical excitation. First, we develop a scalable model of paradoxical excitation in zebrafish. Then, we use this model in large-scale chemical screens to identify dozens of compounds that cause paradoxical excitation. Third, we use these compounds as research tools to identify receptors affecting sedation and paradoxical excitation. Finally, we map whole-brain activity patterns during these behavioral states. Together, these studies improve our understanding of how small molecules cause sedation and paradoxical excitation and may help to accelerate the pace of CNS drug discovery.

## Results

**GABA$_A$R ligands produce paradoxical excitation in zebrafish.** To determine how sedatives affect zebrafish behavior, we assembled a set of 27 CNS depressants in ten functional classes (Fig. 1a, Supplementary Table 1) and tested these compounds in a battery of automated behavioral assays. The behavioral assays were originally devised to discriminate between a broad range of neuroactive compounds[23]. Here, the assays were used to profile anesthetics and other CNS-depressants. In one assay, we used excitatory violet light stimuli to identify compounds that reduce motor activity (Fig. 1b, Supplementary Movie 1). In another assay, we used low-volume acoustic stimuli to identify compounds that enhance startle sensitivity (Fig. 1b, Supplementary Movie 2). Most CNS depressants caused a dose-dependent decrease in animals' average motion index (MI) (Supplementary Fig. 1), however, we noticed a striking exception.

Two anesthetic GABA$_A$R ligands, etomidate and propofol, caused enhanced acoustic startle responses (eASRs). These eASRs occurred in response to a specific low-volume acoustic stimulus, but not to other stimuli (Fig. 1a, b, Supplementary Figs. 2 and 3). Unlike vehicle-treated controls, all the animals in a well treated with etomidate showed robust eASRs (Supplementary Movie 3 and 4). High speed video revealed that the eASRs resembled short latency C-bends (Fig. 1c). Multiple eASRs could be elicited with multiple stimuli (Fig. 1d). Etomidate's half maximal effective concentration (EC$_{50}$) for causing eASRs was 1 μM, consistent with its EC$_{50}$ against GABA$_A$Rs in vitro (Fig. 1e)[27]. Neither etomidate or propofol was toxic at the concentrations that caused eASRs (Supplementary Table 2). The eASRs persisted for several hours and rapidly reversed after drug washout (Fig. 1b, f). Curiously, not all anesthetics caused eASR behaviors in zebrafish, including isoflurane (a volatile inhalational anesthetic that is relatively toxic in zebrafish), dexmedetomidine (a veterinary anesthetic and alpha-adrenergic agonist), and tricaine (a local anesthetic and sodium channel blocker) (Fig. 1a). Together, these data suggest that a subset of GABA$_A$R ligands produce sedation and paradoxical excitation in zebrafish.

To determine if other GABA$_A$R ligands caused similar phenotypes, we used the phenoscore metric to quantify similarities between the archetypal profile caused by etomidate (6.5 uM) and a diverse range of GABAergic compounds (Supplementary Table 3). Average phenoscores of DMSO-treated negative controls were significantly less than etomidate-treated positive controls (0.2 versus 0.71, $P < 10^{-10}$, Kolmogorov–Smirnov test) (Supplementary Fig. 20). Average phenoscores for the test compounds fell on a continuum between the positive and negative controls (Fig. 1g). Based on statistical simulations, these phenoscores were subdivided into three

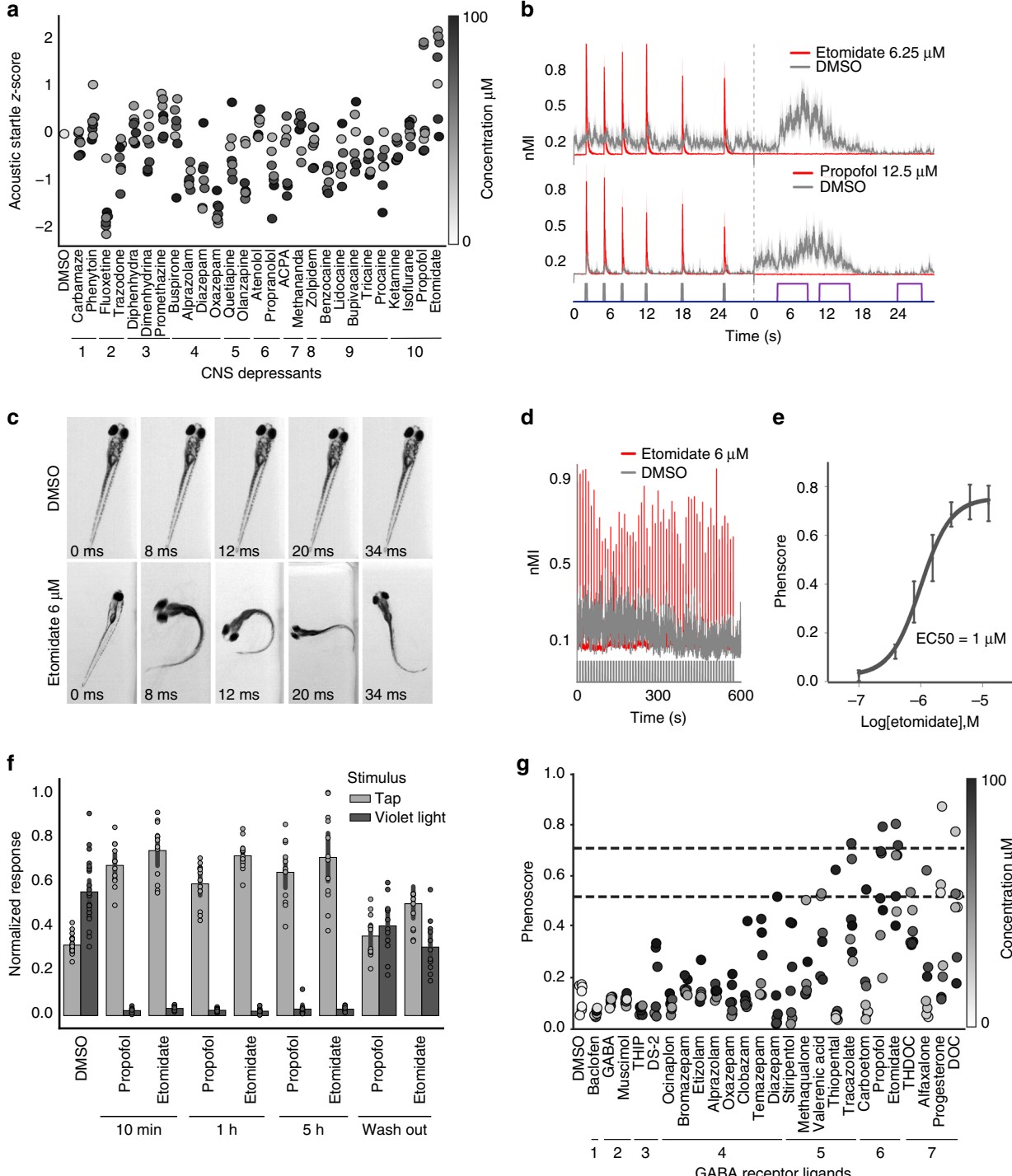

categories: weak, intermediate, and strong (Supplementary Note 1).

Compounds with the strongest phenoscores ($0.71 < x < 1$) included several anesthetic and neurosteroid PAMs including etomidate, propofol, progesterone, and 11-deoxycorticosterone (DOC) (Fig. 1g). The highest scoring treatments for these compounds produced behavioral profiles that were both strongly sedating and produced high-magnitude eASRs (Supplementary Fig. 2). These profiles were not statistically different from the positive controls ($P > 0.05$, Kolmogorov–Smirnov test, Supplementary Fig. 20). Together, these data suggest that a subset of GABA$_A$R PAMs cause sedation and paradoxical excitation in zebrafish. However, due to the overlapping pharmacology of numerous GABA$_A$R subtypes, these data do not clearly point to any specific subset of receptor subtypes as being necessary or sufficient for these behaviors.

In humans, the M-current is a low-threshold, non-inactivating, voltage-dependent current that limits repetitive action potentials and has been implicated in propofol-induced paradoxical excitation[28,29]. To determine if M-currents affect eASRs in zebrafish, we tested several M-current activators and inhibitors. In animals treated with M-current activators (flupirtine[30,31] and ICA-069673[32]), eASR magnitudes significantly decreased (Supplementary Fig. 6, $P < 0.01$, two-tailed $t$-test, $n = 6$ wells; 8 fish/

**Fig. 1** GABA$_A$R PAMs enhance acoustic startle in zebrafish. Zebrafish were treated with the indicated compounds and analyzed for changes in behavioral responses. **a** The scatter plot quantifies acoustic startle response as a z-score (y-axis) in zebrafish treated with the indicated CNS-depressants (x-axis) at the indicated concentrations (colorbar). Each point represents the average of $n = 12$ wells and 6 experimental replicates (also listed in Supplementary Table 1). **b** These plots show how the indicated compounds impact zebrafish motor activity (y-axis) over time (x-axis) ($n = 12$ wells, shaded boundary = 95% confidence interval; nMI, normalized motion index). Colored bars above the x-axis represent the timing and duration of low-volume acoustic stimuli (gray bars) and violet light stimuli (purple bars). The vertical dotted line indicates where the first assay ends and the second begins. **c** Representative images of animals treated with the indicated compounds. Time stamps indicate the time elapsed from the initial presentation of a low-volume acoustic stimulus. **d** These plots compare the motor activity (y-axis) over time (x-axis) of animals treated with DMSO (gray) or etomidate (red) ($n = 50$ larvae). Consecutive stimuli ($n = 60$) are indicated by vertical gray bars. **e** Dose-response curve showing phenoscores at the indicated concentrations (each point represents $n = 12$ wells/dose, error bars: ± SD). **f** Bar plot showing normalized response to the indicated stimulus (tap or violet light) of animals treated with DMSO, 6 μM propofol, or 6 μM etomidate ($n = 12$ wells, error bars: ± SD) for the indicated durations. **g** Average phenoscores (y-axis) of zebrafish treated with the indicated compounds. Dashed lines intersecting the y-axis at 0.51 and 0.71 correspond respectively to 1% and 5% significance cutoffs, as determined from statistical simulations. Compounds are grouped by ligand class: (1) GABA$_B$R agonist, (2) GABA$_A$R orthosteric agonist, (3) PAM of δ-subunit containing GABA$_A$Rs, (4) GABA$_A$R BZ-site PAM, (5) GABA$_A$R non-BZ-site PAM, (6) GABA$_A$R neurosteroid PAM, (7) GABA$_A$R anesthetic PAM

well). By contrast, in animals treated with M-current inhibitors (linopirdine, XE-991, and oxotremorine[33]) eASR magnitudes significantly increased (Supplementary Fig. 6, $P < 0.000001$, two-tailed $t$-test, $n = 6$ wells; 8 fish/well). These data suggest that zebrafish eASRs are a form of paradoxical excitation affected by potassium channel M-currents.

**Large-scale behavioral screening identifies hit compounds**. To prepare for large-scale screening, we calculated phenoscores for hundreds of positive and negative control wells (treated with etomidate or DMSO, respectively). The average phenoscores of the positive controls were significantly greater than the negative controls ($0.7, \pm 0.11$ SD versus $0.1 \pm 0.05$ SD), suggesting that a large-scale screen would have an expected false positive and negative rate of 2% and 0.4%, respectively (at a threshold of 3 SD) (Fig. 2a, Z-factor = 0.7, $n = 944$ wells).

Then, we screened a library of 9512 structurally-diverse compounds plus 2336 DMSO-treated negative controls, and analyzed their effects on sedation and paradoxical excitation. Visualized as a contour plot, the highest density of phenoscores occurred in three major regions (Fig. 2b). The first region contained 11,679 compounds and DMSO-treated control wells that did not phenocopy etomidate. The second region contained 44 potentially toxic compounds that immobilized zebrafish but did not cause paradoxical excitation. The third region contained 125 compounds that both produced immobilization and phenocopied etomidate and were considered primary screening hits (Supplementary Table 4, Supplementary Fig. 7).

To organize these hit compounds by structural similarity, we clustered them based on Tanimoto similarities and visualized the results as a dendrogram that contained 14 clusters (Fig. 2c). Several clusters included compounds previously associated with GABA$_A$Rs (Fig. 2c, d). For example, Cluster 10 included several dihydro/quinazolinones that are structurally-related to methaqualone, a sedative hypnotic drug (Fig. 2c, d). A second cluster, Cluster 14, included several isoflavonoids, which are structurally-related to flavonoid sedatives[34]. Overall, we selected a broad range of 57 primary hit compounds across all the clusters to re-purchase and re-test (Supplementary Table 4). Each compound was re-tested in a dose-response format from 0.1 to 100 μM and scored based on its ability to immobilize zebrafish and increase eASRs. Together, 81% of these primary hit compounds (46/57) caused reproducible eASR phenotypes at one or more concentrations (Fig. 2e, Supplementary Fig. 8, and Supplementary Table 4), indicating a high rate of reproducibility from the primary screen.

To determine if these compounds targeted human GABA$_A$Rs, we tested them in a fluorescent imaging plate reader (FLIPR) assay on HEK293 cells transfected with $\alpha_1\beta_2$ and $\alpha_1\beta_2\gamma_2$ human GABA$_A$R subtypes. In this cell line, etomidate, tracazolate, and

propofol increased fluorescence in the presence of GABA, as expected for GABA$_A$R PAMs. In addition, half of the tested hit compounds (23/46) also showed PAM activity (Fig. 2f, Supplementary Fig. 9). By contrast, PAM activity was not observed with negative control compounds including BGC 20-761 (an HTR6 antagonist) and PTX (a GABA$_A$R channel blocker) which likely reduced GABA$_A$R activity due to inhibition of constitutively active GABA$_A$Rs in the system. Interestingly, the PAM activity of two hit compounds, 7013338 and 5942595, was significantly greater than the positive controls (Fig. 2f, $P < 0.0001$, two-tailed $t$-test, $n = 4$). While some of the compounds appeared to function in this assay as negative allosteric modulators (NAMs), reductions in fluorescence were likely due to toxicity-induced cell loss (Fig. 2f, Supplementary Table 5). In addition to the PAM assay, four hit compounds directly activated GABA$_A$Rs in the absence of GABA, including 5860357, 6091285, 5835629, and 7284610 (Supplementary Fig. 9). The strongest direct activator, 5835629, did not further enhance GABA$_A$R activation in the PAM assay, presumably because the cells were already maximally activated by the compound before GABA was added. These data suggest that behavioral screens in zebrafish can enrich for compounds with activity at specific human receptors. In addition, these data suggest that many of the hit compounds identified in the screen cause sedation and paradoxical excitation via GABA$_A$Rs.

**Hit compounds act on targets including GABA$_A$R and HTR6**. To determine if any of the hit compounds acted on non-GABA$_A$R targets, we used the Similarity Ensemble Approach (SEA)[35] to predict targets based on 'guilt-by-association' enrichment factor scores (EFs). Among the top-ranked 1000 screening compounds, 150 targets were enriched (Supplementary Table 6). As we analyzed subsets of hit compounds with increasing phenotypic stringency, the number of enriched targets decreased (Fig. 3a, b). The most stringent set of 30 top-ranked hit compounds contained 25 enriched targets including GABA$_A$Rs, 5α-reductase, mGluRs, and 5-HTRs (Supplementary Table 7). By contrast, this stringent set of hit compounds were not enriched for other sporadically predicted targets such as histone deacetylase, matrix metalloproteinase, and carbonic anhydrase (Fig. 3b). As additional negative controls, we tested 48 reference compounds targeting receptors with relatively low EF scores and confirmed they did not cause eASR phenotypes at any concentration (Supplementary Table 8, Supplementary Fig. 10). Together, these data suggest that the hit compounds may act on GABA$_A$Rs, 5α-reductase, mGluRs, or 5-HTRs (Supplementary Note 2).

A second approach to target identification was to test the binding affinity of hit compounds against a panel of 43 human and rodent neurotransmitter-related targets. Of 46 hit compounds, 33 of them bound to at least one of 19 receptors at a $K_i <$

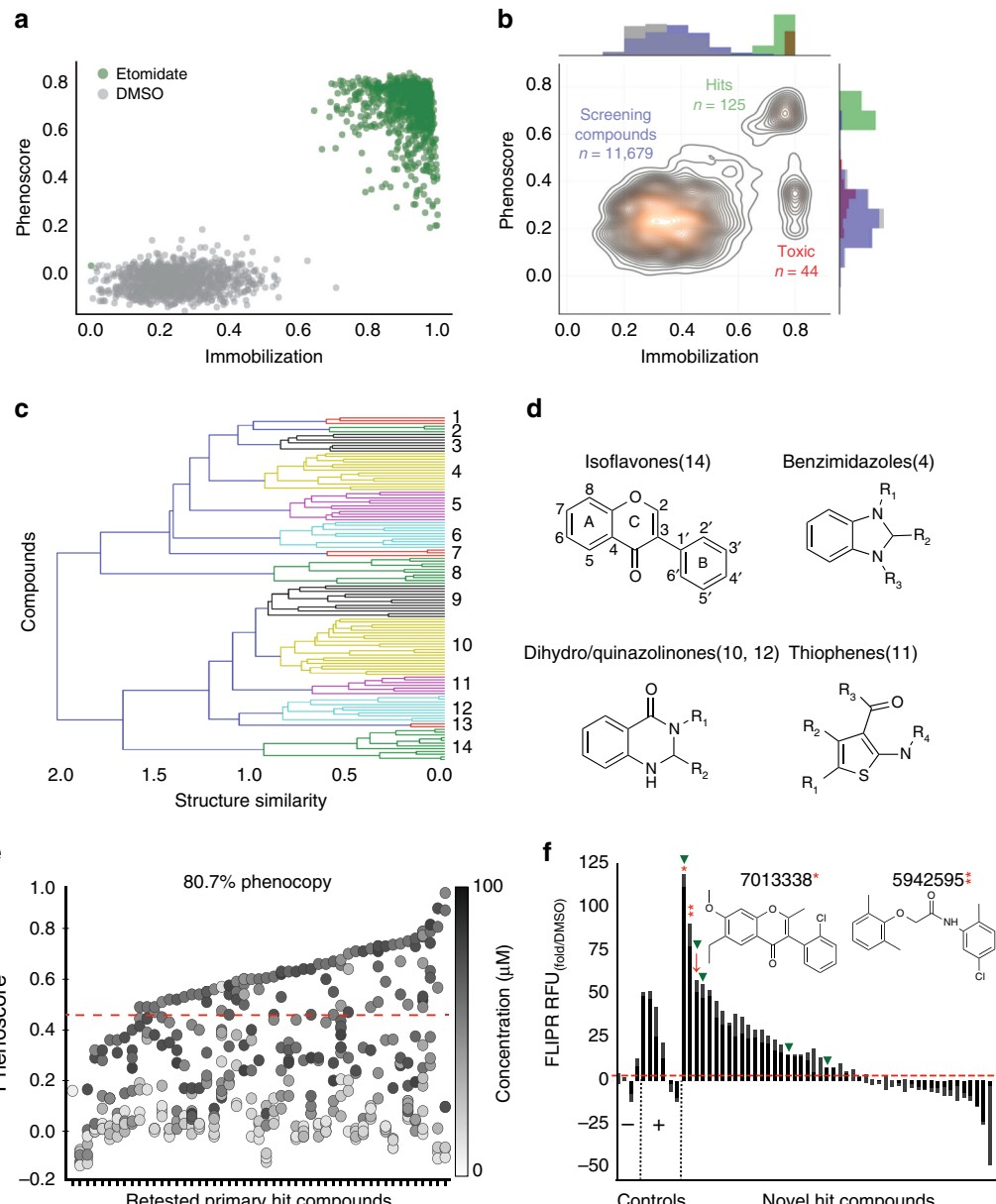

**Fig. 2** A high-throughput behavioral screen identifies GABAergic compounds. Zebrafish were treated with various compounds and analyzed for anesthetic-related behaviors. **a** This scatter plot compares phenoscores of individual wells treated with DMSO or etomidate (6.25 μM) (Z-factor = 0.7, n = 944 wells). **b** This contour plot scores each well from the large-scale behavior-based chemical screen (11,679 compounds, 2336 DMSO controls) by its phenoscore (y-axis) and immobilization index (x-axis). Labels indicate regions with 125 hit compounds (green), 44 toxic compounds (red), and the remaining screening compounds and DMSO controls (blue and gray, respectively). **c** Structural clustering of the top 125 hit compounds (y-axis) forms 14 clusters using a Tanimoto similarity metric (x-axis). **d** Example structures of selected compounds in the indicated clusters. **e** This scatter plot shows a 80.7% reproducibility rate for 57 primary hit compounds. Each point represents the average phenoscore of n = 12 wells at the indicated concentrations (colorbar). The first column represents DMSO controls; the order of other compounds are listed in Supplementary Table 4. **f** Human GABA$_A$R activation (y-axis) was measured by FLIPR analysis. Of 47 hit compounds, 23 potentiated GABA$_A$Rs. Compounds 7013338 and 5942595 potentiated GABA$_A$Rs significantly greater than positive controls (red asterisk = 7013338, two red asterisks = 5942595, P < 0.0001, two-tailed t-test, n = 2–4 replicates as indicated). The hit threshold was defined as 2× the average DMSO control group. Picrotoxin, BGC 20-761, progesterone, and DMSO were used as negative controls (x-axis) while etomidate, tracazolate, propofol, diazepam, and thiopental were used as positive controls. Arrows indicate compounds that were predicted by SEA to bind GABA$_A$Rs (red arrows) and compounds that bound to TSPO in vitro (green arrowheads)

10 μM (Fig. 3c, d). Several hit compounds bound to multiple targets, including compound 7145248, which bound to TSPO, the benzodiazepine receptor (BZP), dopamine transporter, and the alpha 2b receptor (Fig. 3d). The most common targets (binding > 5 compounds) included BZP, sigma 2, HTR2A/B/C, HTR6, and TSPO (Fig. 3c, d). TSPO, previously known as the peripheral benzodiazepine receptor (PBR) (because it was originally identified as a binding site for benzodiazepine anxiolytic drugs), is a mitochondrial protein that supplies cholesterol to steroid-producing enzymes in the brain[36]. TSPO ligands are thought to enhance GABAergic signaling by increasing neurosteroid production. However, some TSPO ligands, including

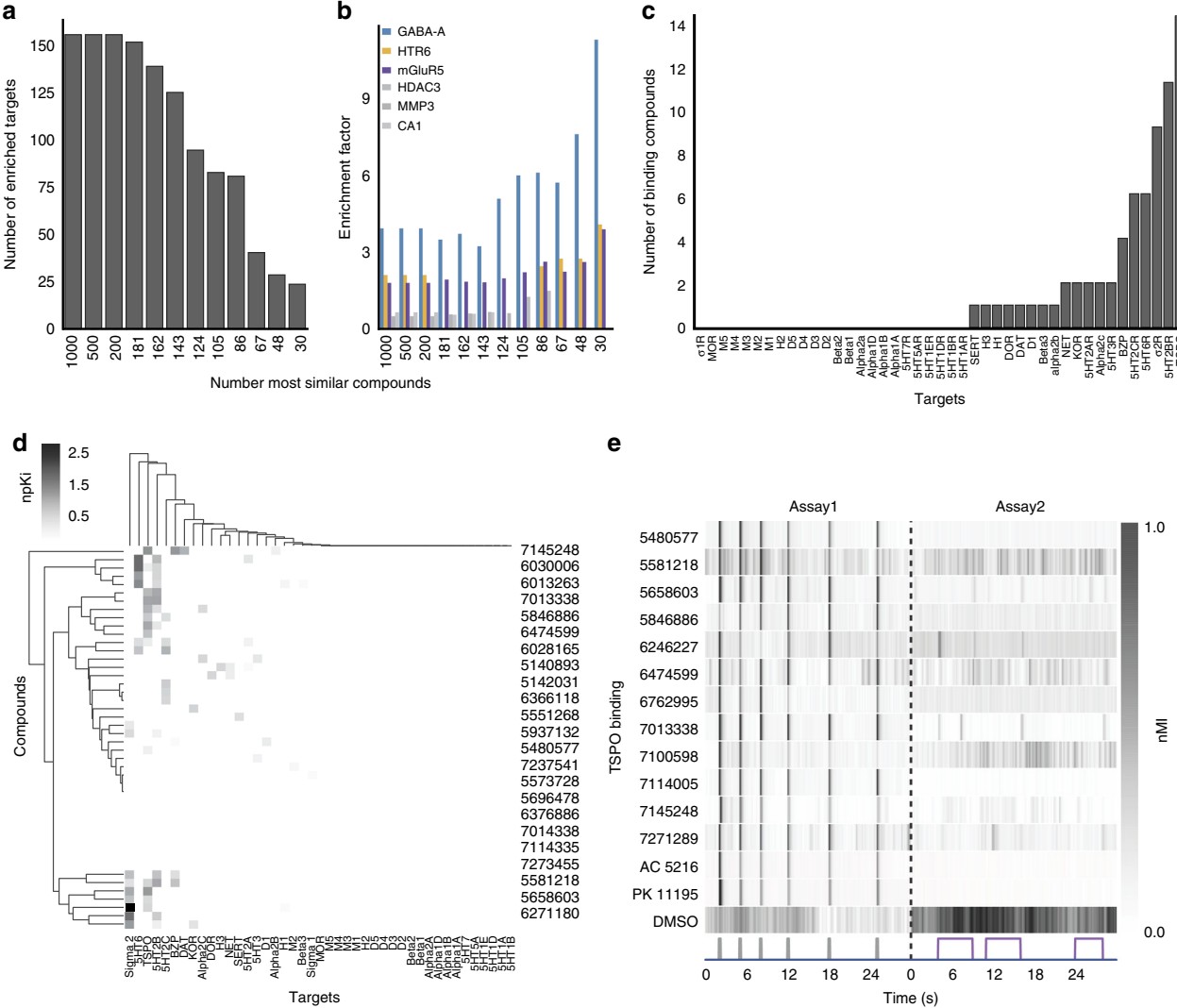

**Fig. 3** Potential targets include GABA$_A$R, mGluR, TSPO, and HTR6. **a** SEA analysis was performed on decreasing numbers of hit compounds (1000-30). The bar plot shows the number of SEA enriched targets decreasing as the analysis focuses on the top 30 hit compounds. **b** The bar plot shows increasing enrichment of GABA$_A$R, HTR6, and mGluR5 as the top targets predicted for the top 30 hit compounds. **c** This bar plot shows the number of 46 primary hit compounds (y-axis) that bound to the indicated CNS receptors (x-axis). **d** The clustergram shows binding affinity profiles at the indicated CNS receptors. The colorbar indicates normalized Ki (npKI). **e** Heatmap of average motor activity profiles for TSPO binding compounds (y-axis) over time (x-axis) (n = 12 wells). Assay 1 is comprised of 6 low-amplitude acoustic stimuli (gray); Assay 2 is a series of 3 violet light pulses (violet). These two assays are separated by a dotted line. AC 5216 and PK 11195 are TSPO binding compounds. Abbreviations: nMI, normalized motion index; MMP3, matrix metallopeptidase 3; CA1, carbonic anhydrase 1; HDAC3, histone deacetylase 3

benzodiazepines and zolpidem, also bind to GABA$_A$Rs directly[37]. We found that 14 hit compounds bound to TSPO in vitro (Fig. 3c, d), and that two TSPO reference ligands, PK 11195 and AC 5216, both caused eASRs in vivo (Fig. 3e). Of the 14 compounds that bound to TSPO in vitro, 5 compounds potentiated GABA$_A$R in FLIPR assays (Fig. 3c; 2f, green arrowheads). These data suggest that TSPO ligands promote anesthetic-related phenotypes via direct interactions with GABA$_A$Rs, indirect effects on neurosteroidogenesis, or both.

Both target identification approaches, SEA and the in vitro receptor binding assays, implicated HTR6. For example, SEA predicted that seven hit compounds, six benzenesulfonamides and one piperazine, targeted HTR6 (Fig. 4a, Supplementary Table 9). These compounds reproducibly caused eASRs in vivo (Fig. 4b). In vitro, six of these hit compounds bound to HTR6 at nanomolar concentrations (Ki = 54–807 nM) (Fig. 4c). To

determine their functional effects, we tested them for agonist and antagonist activity in G-protein and β-arrestin pathways at eight serotonin receptor subtypes (1A, 2A, 2B, 2C, 4, 5A, 6, and 7A). Six of the compounds antagonized HTR6 in vitro. Most of them antagonized both G-protein and β-arrestin pathways, suggesting that the compounds were unbiased HTR6 antagonists (activity range 3.30 nM–18.2 μM) (Fig. 4d). By contrast, a structurally-related piperazine, 5801496, did not cause eASRs in vivo. To determine if any annotated HTR6 antagonists also caused eASRs, we analyzed six structurally-diverse HTR6 reference antagonists. Two of these reference antagonists, BGC 20-761 and idalopirdine, reproducibly caused eASRs in vivo (Fig. 4b). It is unclear why only 2/6 reference HTR6 antagonists caused eASRs in zebrafish, but it may be related to issues with absorption, metabolic stability, and/or structural differences between human and zebrafish receptors. A panel of 36 additional

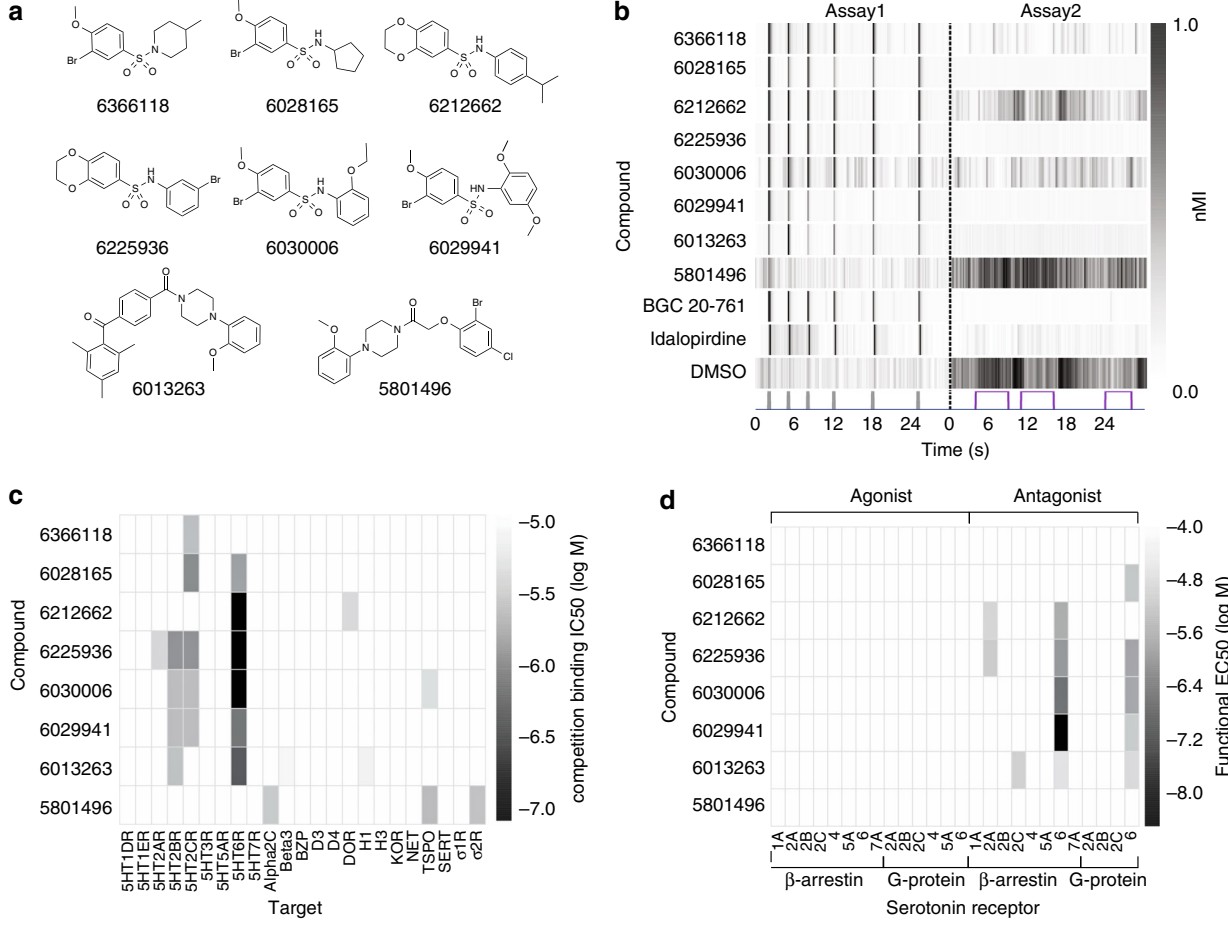

**Fig. 4** A subset of hit compounds are HTR6 antagonists. **a** Structures of eight primary hit compounds predicted to bind HTR6. **b** Heatmap showing average ($n = 12$) motor activity profiles over time (x-axis) for compounds predicted to bind HTR6 (y-axis). Assay 1 is comprised of 6 low-amplitude acoustic stimuli; Assay 2 is a series of 3 violet light pulses (as indicated), these two assays are separated by a dotted line. BGC 20-761 and Idalopirdine are previously annotated HTR6 antagonists. **c** Heatmap showing binding affinities of primary hit compounds at 23 CNS receptors (x-axis). **d** Heatmap showing functional activity of primary hit compounds at the indicated GPCRs. (nMI, normalized motion index)

5-HT modulating ligands at various serotonergic targets did not cause eASRs at any concentration tested (Supplementary Table 9). Together, these data suggest that a subset of HTR6 antagonists cause eASRs in zebrafish.

**A neural substrate for paradoxical excitation**. To determine which regions of the brain were active during eASRs, we visualized whole-brain activity patterns by pERK labeling[38]. In DMSO-treated control animals, pERK labeling showed broad patterns of activity in the optic tectum, telencephalon, and other brain regions (Fig. 5a, b). By contrast, in animals treated with etomidate or propofol, pERK labeling was broadly reduced (Fig. 5c–e; $P < 0.0005$, Mann–Whitney $U$ test). Acoustic stimuli significantly activated a cluster of neurons in the caudal hindbrain at the base of the 4th ventricle near the auditory brainstem and the nucleus of the solitary tract (NST)[39] at the level of the area postrema (Fig. 5f, g; $P < 0.0005$, Mann–Whitney $U$ test)[40], suggesting that this hindbrain neuron cluster represented a specific substrate of eASR behavior.

To determine if activity in this region specifically occurred during eASRs, we analyzed pERK labeling in this region during four other robust motor behaviors. First, in animals stimulated by optovin (a reversible photoactivatable TRPA1 ligand)[24], neuronal activity increased in many brain regions including the

telencephalon and optic tectum but not in the hindbrain (Fig. 5j, k). Second, in DMSO-treated control animals stimulated by light, neuronal activity increased in the telencephalon and pineal gland but not in the hindbrain (Supplementary Fig. 13a). Third, in animals stimulated by the GABA$_A$R antagonist picrotoxin (PTX), neuronal activity increased in the telencephalon but not in the hindbrain (Supplementary Fig. 13b). Finally, in DMSO-treated animals stimulated by a strong acoustic stimulus (hard tap), pERK labeling increased in the midbrain but not in the caudal hindbrain (Supplementary Fig. 13c). Compared to lower concentrations of etomidate (6 μM), higher concentrations of etomidate (50 μM) suppressed eASRs and decreased pERK labeling in the caudal hindbrain (Supplementary Fig. 13d). Like etomidate, BGC 20-761 reduced neuronal activity throughout most of the brain and increased activity in the caudal hindbrain neuron cluster in response to acoustic stimuli (Fig. 5h, i, Supplementary Note 3). Together, these data suggest that the labeled neurons in the caudal hindbrain are a specific substrate of eASR behaviors.

**Hit compounds produce distinct side effect profiles**. To prioritize hit compounds for further development, we tested them for specific side effects. For example, a serious side effect of etomidate is that it suppresses corticosteroid synthesis due to

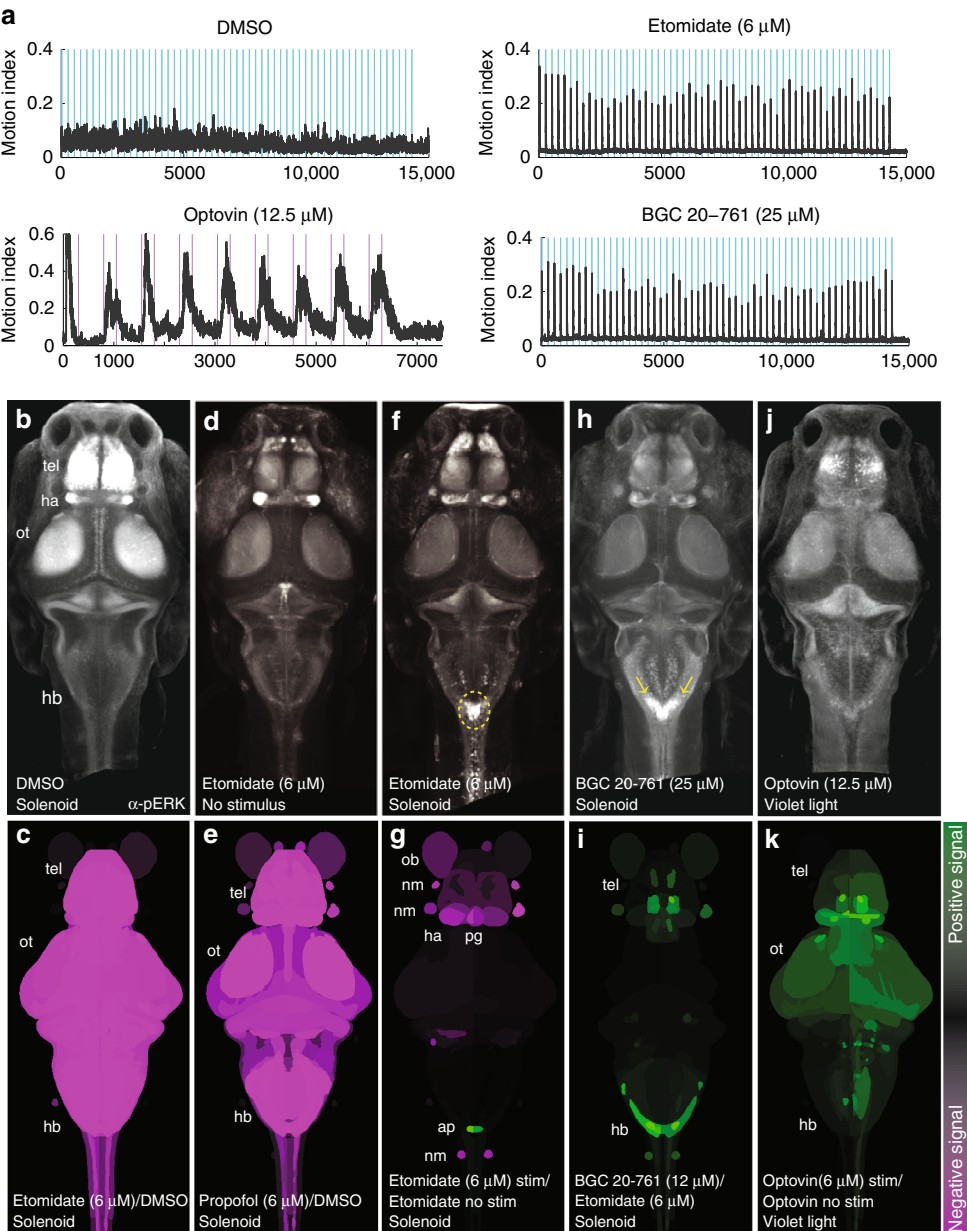

**Fig. 5** Hit compounds activate hindbrain neurons. Animals were exposed to the indicated drugs and stimuli and analyzed for pERK levels as a readout of neuronal activity. **a** Plots showing motor activity (y-axis) over time (x-axis) for animals treated with the indicated compounds ($n = 25–50$ larvae) in response to the indicated acoustic (blue) or violet light (purple) stimuli. **b, d, f, h, j** Confocal projections showing the average fluorescent intensity of image registered larval brains stained with α-pERK ($n = 10$ larvae/condition). Larvae were treated with the indicated compounds and exposed to the low-amplitude acoustic stimulus once every 10 s for 10 min, except for (**b**, no stimulus) and (**f**, violet light exposure). **c, e, g, i, k** Brain activity maps showing significant ΔpERK signals using the Z-brain online reference tool ($n = 5–10$ animals/condition). The heatmap indicates positive (green), negative (purple), and nonsignificant (black) changes in pERK labeling ($P < 0.0005$, Mann–Whitney $U$ test). All activity maps are comparisons between the indicated treatment conditions. Abbreviations: tel, telencephalon; ot, optic tectum; hb, hindbrain; ob, olfactory bulb; nm, neuromast; ap, area postrema; pg, pineal gland

off-target activity on 11β-hydroxylase, the enzyme that synthesizes cortisol in humans and zebrafish. To determine if any of the hit compounds suffered from similar liabilities, we measured their effects on cortisol levels. As a positive control, we used carboetomidate, a close structural analog of etomidate that was rationally designed to retain etomidate's activity on GABA$_A$Rs, while disrupting its ability to suppress cortisol synthesis. Both etomidate and carboetomidate immobilized zebrafish and caused eASRs (Fig. 1g). As expected, etomidate

lowered cortisol levels in zebrafish, whereas carboetomidate did not, suggesting that these compounds have similar side effects in humans and zebrafish (Fig. 6a). Next, we tested 12 hit compounds in the cortisol assay, including eight GABA$_A$R ligands (found to be positive in the FLIPR assay), one compound predicted to target GABA$_A$R by SEA (5951201), two HTR6 antagonists (6225936 and 6029941), and one mysterious compound with no target leads (5736224). None of these compounds reduced cortisol levels in zebrafish (Fig. 6a),

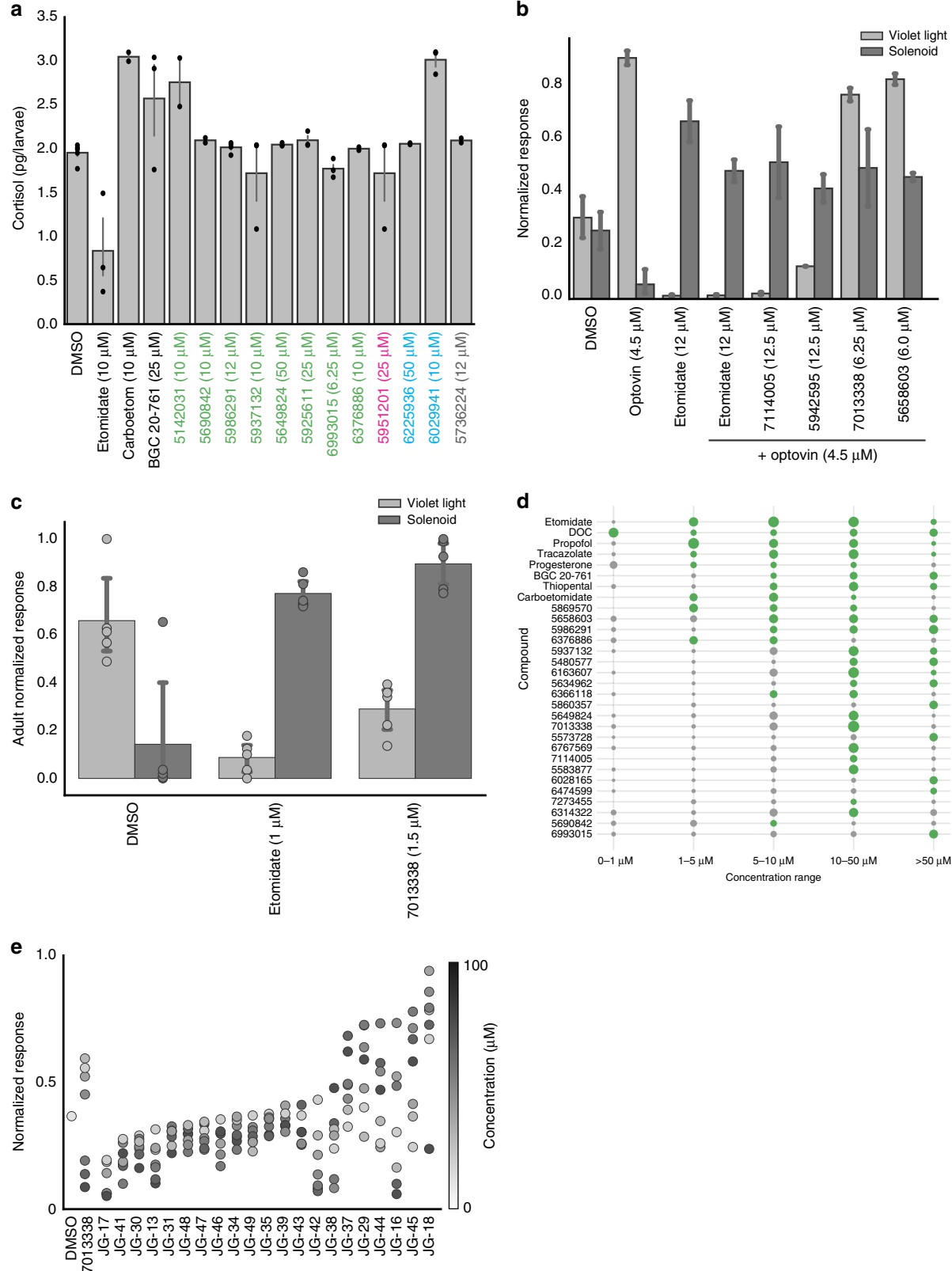

indicating that these ligands cause sedation and paradoxical excitation without suppressing cortisol levels.

To determine if any of the hit compounds may be analgesic, we used optovin-induced motor activity as a potential analgesia-related assay. In humans, general anesthetics reduce perceptions of pain and suffering. Although it is unclear if fish feel pain, painful stimuli in humans also cause behavioral responses in zebrafish. For example, activation of the TRPA1 ion channel causes pain in humans[41], and optovin, a photoinducible TRPA1 ligand, induces strong behavioral responses in zebrafish[24]. As a

**Fig. 6** Hit compounds show diverse efficacy windows and side effect profiles. **a** This bar plot shows cortisol levels (y-axis) in animals treated with the indicated compounds (x-axis) including FLIPR-positive GABAergics (green), SEA predicted GABAergics (magenta), serotonergics (blue), and a compound with undetermined targets (gray) (n = 2–5 experiments, 15 animals/experiment, error bars: ± SEM). **b** This bar plot shows the normalized responses (y-axis) of animals treated with the indicated compounds (x-axis) in the pain-related optovin-response suppression assay. **c** This bar plot shows the magnitude of behavioral responses of adult zebrafish (y-axis) treated with with the indicated compounds (x-axis). **d** Dot plot showing efficacy windows for the indicated compounds with strong (green) or weak (gray) phenocopy scores. Marker size represents the magnitude of the eASR response (n = 12 wells/condition). Compounds with broad efficacy windows have large green dots at multiple concentrations (x-axis). **e** This strip plot shows the normalized acoustic startle response (y-axis) of larvae treated with increasing concentrations (colorbar) of multiple analogs of the screening hit 7013338 (x-axis) (n = 4–6 wells/condition, 8 fish/well)

positive control, we found that etomidate suppressed the optovin response at the same concentrations that caused eASRs (Fig. 6b). Similarly, we found that two GABAergic hit compounds, 7114005 and 5942595, also blocked optovin-induced motor activity at the same concentrations that caused eASRs (Fig. 6b). By contrast, compounds 5658603 and 7013338 did not suppress the optovin response (Fig. 6b). The HTR6 antagonist BGC 20-761 also blocked the optovin response (Supplementary Fig. 16), however serotonergic hit compounds 6225936, 6028165, and 6212662 only reduced the optovin response at concentrations that also reduced eASRs (Supplementary Fig. 16). Together, these data suggest that the mechanisms controlling sedation and eASRs may be separable from analgesia, and that some eASR-causing compounds may cause analgesic-related effects in zebrafish.

To determine if eASRs also occur in adult zebrafish, we treated adult animals with etomidate and the hit compound 7013338, the most effective hit compound in the FLIPR assays (Fig. 2f). We found that both of these compounds also worked in adult animals, reducing the violet light response, while increasing acoustic startle (Fig. 6c). These data suggest that the mechanisms underlying eASR phenotypes are not limited to larvae but also exist in adult zebrafish.

In humans, therapeutic windows for many inhalational anesthetics are only 2-fold, while therapeutic indices for intravenous anesthetics are not much better[42]. Many of the hit compounds also had relatively narrow efficacy windows (Fig. 6d). Numerous analogs of key hit compounds including thiophenes, aryloxycarboxamides, quinazolines, and sulfonamides had lower activity than the original hits (Supplementary Fig. 17), suggesting that substantial medicinal chemistry would be needed to increase the potency of the primary hit compounds.

Compound 7013338 activated human $GABA_ARs$ more than any other hit compound in the FLIPR assay (Fig. 2f). However, its efficacy window was relatively narrow (10–50 μM), raising questions about its structure activity relationship (SAR) (Fig. 6e). To analyze its SAR, we generated 21 analogs with different substituents on the A-, B-, and C-rings (Fig. 2d, Supplementary Fig. 18, and Supplementary Table 10), and tested these analogs for biological activity in vivo. The most active analog, JG-18, increased eASR magnitude and widened the efficacy window from 1 to 50 μM (Fig. 6e). It had a chloro substituent on C2' of the B-ring, an ethyl substituent on C2 of the C-ring, and C-6 propyl and C7 hydroxyl substituents on the A-ring (Supplementary Fig. 18). In congruence with previous SAR analyses of isoflavones[34], JG-18 and other analogs with more lipophilic substituents on position C6 of the A-ring and position C2 of C-ring exhibited increased biological activity (Fig. 6e). By contrast, analogs with more dramatic enhancements in steric bulk and lipophilicity at these positions (i.e., phenethyl and propyl, respectively) exhibited reduced biological activity. Likewise, it was notable that capping the polar C7-hydroxy group of JG-18 with alkyl and acyl groups tended to lessen biological activity. Importantly, we found that the B-ring C2' chloro substituent was absolutely critical for biological

activity, since analogs without it did not cause eASRs. Previously, it was reported that analogs with alkoxy or trifluoromethoxy substituents at multiple positions but especially at C3' on the B-ring were high affinity $GABA_AR$ binders in vitro[43]. Surprisingly, compound JG-17 (with a trifluoromethoxy substituent on C3' of the B-ring, an ethyl substituent on C2 of C-ring, and C6 propyl and C7 hydroxy substituents on the A-ring) had no biological activity in vivo (Fig. 6e). It is not clear why these ligands were not active in zebrafish. Perhaps, the anomaly could be ascribed to low penetrance in vivo, receptor subtype selectivity, and/or structural differences between the human and zebrafish $GABA_ARs$. Together, these data suggest that additional SAR analyses may yield analogs with greater potency and broader efficacy windows in vivo.

## Discussion

These studies show that anesthetics and other $GABA_AR$ PAMs cause sedation and paradoxical excitation in zebrafish, and that this behavioral model has high predictive and construct validity for identifying modulators of human $GABA_ARs$. Indeed, these studies may have underestimated the number of hit compounds that targeted $GABA_ARs$ for several reasons. One reason is that the in vitro $GABA_AR$ FLIPR assay only tested a very small number of receptor subtypes and subunit isoforms ($\alpha_1\beta_2$ and $\alpha_1\beta_2\gamma_2$). As a result, these assays would have missed compounds that acted on other $GABA_AR$ subtypes. A second reason is that some of the hit compounds may act on zebrafish-specific $GABA_ARs$. Finally, some hit compounds that caused eASRs in zebrafish may need to be bioactivated in vivo, and would therefore not be be active in vitro. Therefore, even more of the hit compounds may have targeted $GABA_ARs$.

These studies also suggest that non-$GABA_AR$ mechanisms may also affect paradoxical excitation, including HTR6 antagonism. For example, we found that HTR6 antagonists produced sedation and paradoxical excitation in zebrafish (Fig. 4). These HTR6 antagonists likely reduce neuronal excitation via different mechanisms than $GABA_AR$ PAMs. $GABA_ARs$ are widely distributed in the CNS, suggesting that GABA ligands likely inhibit most neurons directly. By contrast, HTR6s are restricted to discrete neuronal populations[44], suggesting that their effects are likely propagated through indirect signaling networks. HTR6 antagonists can reduce 5-HT neuronal firing[45], presumably by blocking positive feedback[46] control of raphe neurons that broadly project throughout the brain and spinal cord[21]. Researchers have made remarkable progress applying the principles of systems pharmacology to structure-based target predictions[47], computer assisted design of multi-target ligands[48–50], and the large-scale prediction of beneficial drug combinations[10,51]. Although we focused on predicting targets of compounds one at a time, in future studies it may be possible to calculate multi-target enrichment factors among the hit compounds from large-scale phenocopy screens and identify multi-target mechanisms.

The HTR6 antagonists identified in this study add to a growing body of evidence implicating various serotonin ligands and receptors in phenotypes related to neuronal inhibition and excitation. Our finding that HTR6 antagonists activate a region in the zebrafish NST (Fig. 5h, i), are consistent with previous work showing that HTR6 antagonists activate neurons in the mammalian NST[46]. In rodents, HTR6 antagonists promote sleep[47], reduce anxiety[48], and show anticonvulsant properties[49]. However, it is not clear if HTR6 causes these effects via specific neuronal circuits, or more generally by coordinating nervous system tone and arousal. Furthermore, there are substantial differences in the central nervous system distribution and pharmacology of the mouse, rat, and human HTR6 receptors[50]. So, although HTR6 antagonists phenocopied etomidate in zebrafish, these effects may not translate to anesthetic activity in humans. Despite promising effects in rodents, several HTR6 antagonists failed in clinical trials as cognitive enhancers for the treatment of Alzheimer's disease[51], underscoring the caveats of generalizing between humans and model organisms.

These data suggest at least two possible models by which GABA$_A$R PAMs could cause paradoxical excitation of the acoustic startle response. One possibility is that the ligands disinhibit the acoustic startle neurons. Alternatively, the ligands may excite specific neurons directly, due to conditions that reverse the chloride equilibrium potential, such as the tonic activation of GABA$_A$Rs[52]. Our observation that caudal hindbrain neurons were activated by acoustic stimuli in etomidate-treated zebrafish is not the first to link GABA signaling to auditory excitation. For example, in rodents, gaboxadol activates extrasynaptic GABA$_A$Rs, hyperpolarizes resting membrane potential, and converts neurons in the auditory thalamus to burst mode[53]. In addition, etomidate causes purposeless muscle excitement that is exacerbated by acoustic stimuli in dogs[54]. In zebrafish, researchers have found that the offset of optogenetic-induced inhibition of caudal hindbrain neurons triggers swim responses[55]. In addition, zebrafish caudal hindbrain neurons have been shown to be activated during hunting behaviors, a behavior that requires strong inhibitory control[56]. However, exactly how these neurons impact motor activity, and why startle neurons remain active despite elevated inhibitory tone, remains unclear.

Although these studies show that GABA$_A$R PAMs cause paradoxical excitation, pharmacological experiments to determine which GABA$_A$R subtypes caused eASRs were ultimately inconclusive. While the majority of GABA$_A$Rs in the CNS are benzodiazepine-sensitive γ-containing subtypes, and multiple benzodiazepines did not cause strong eASRs (Fig. 1g), γ-containing subtypes may still be very important for eASRs. One reason is that the benzodiazepines tested in this study only represent a very small subset of benzodiazepine analogs. Another reason is that diazepam produced intermediate eASR phenotypes (Fig. 1g, Supplementary Fig. 2), suggesting that other benzodiazepines may cause even stronger eASR phenotypes. Although etomidate, propofol, neurosteroids, and other anesthetics are PAMs at δ-subunit containing GABA$_A$R subtypes, these ligands also modulate γ-containing subtypes. Furthermore, although THIP and DS2 are reported to have preferential activity at δ-containing GABA$_A$Rs, these compounds also modulate γ-containing receptors[57], and they did not cause eASRs. One alternative explanation is that β-isoforms[58,59] could drive the presence or absence of eASRs. Another possible explanation is that whereas PAMs may produce immobilizing effects effects via some receptor subtypes, they may produce eASRs via other subtypes. In summary, although a subset of GABA$_A$R PAMs caused eASRs, these compounds may do so via a variety of receptor subtypes. In future studies, it would be interesting to test additional benzodiazepines for such effects including midazolam,

which causes paradoxical excitation in humans[60]. The specificity of currently available pharmacological tools may be insufficient to determine which GABA$_A$R subtypes produce eASRs. Therefore, future studies may require targeted knockouts and other genetic tools to help identify the key receptor subtypes.

While these studies focused on behavioral profiling, other types of phenotypic profiling data may further improve the accuracy of neuroactive compound classification, including whole-brain imaging. Whole-brain imaging allows researchers to record real-time firing patterns will likely add massive amounts data to the behavioral pharmacology field[61,62]. For example, recent advances in high-throughput brain activity mapping for systems neuropharmacology illustrate how whole-brain activity mapping can be used in primary screening for compounds that activate specific circuits, or allow researchers to discriminate between compounds with similar behavioral phenotypes but that work on different neuronal populations[62]. These approaches enable primary screening for compounds that activate specific circuits and allow researchers to discriminate between compounds with similar behavioral phenotypes but that work on different neuronal populations.

In summary, we have shown that GABA$_A$R PAMs cause sedation and paradoxical excitation in zebrafish. Whereas previous behavior-based chemical screens in zebrafish have identified neuroactive compounds related to behaviors including sleep[25], antipsychotics[23], learning[26], and appetite[63], we show here that behavioral profiling can also be used to rapidly identify compounds related to sedation and paradoxical excitation. Future studies will likely expand the utility of behavior-based chemical phenocopy screens to additional kinds of neuroactive ligands, targets, and pathways.

## Methods

**Fish maintenance, breeding, and compound treatments.** We collected a large number of fertilized eggs (up to 10,000 embryos per day) from group matings of wild-type zebrafish (from Singapore). All embryos were raised on a 14/10-hour light/dark cycle at 28 °C until 7 dpf. Larvae were distributed 8 animals per well into square 96-well plates (GE Healthcare Life Sciences) with 300 μL of egg water. Compound stock solutions were applied directly to the egg water and larvae were incubated at room temp for 1 h before behavioral analysis. To determine the impact of group size on this assay, we analyzed eASR behaviors from animals in different group sizes (1, 2, 4, 8, 12, 16, 32 animals per well). Although animals in all groups responded similarly to the stimulus (Supplementary Movie 5), the largest differences between treated and controls were seen in groups of 8 and 16 animals (Supplementary Fig. 19). We, therefore, used 8 animals to balance small group size with a strong MI signal. All zebrafish procedures were and approved by the UCSF's Institutional Animal Care Use Committee (IACUC), and in accordance with the Guide to Care and Use of Laboratory Animals (National Institutes of Health 1996) and conducted according to established protocols that complied with ethical regulations for animal testing and research.

**Compounds and chemical libraries.** All chemical libraries were dissolved in DMSO. The Chembridge library (Chembridge Corporation) contains 10,000 compounds at 1 mM. The Prestwick library (Prestwick Chemical) contains 1280 approved drugs at 10 mM. All compounds were diluted in E3 buffer and screened at 10 μM final concentration in < 1% DMSO. Controls were treated with an equal volume of DMSO. Compounds were validated in 3-12 replicate wells, on 3 replicate plates. For dose-response behavioral assays, compounds were tested at 7 concentrations that ranged from 0.1 to 100 μM, unless otherwise indicated.

**Automated behavioral phenotyping assays.** Digital video was captured at 25 frames per second using an AVT Pike digital camera (Allied Vision). Each assay duration was 30–120 s, and consisted of a combination of acoustic and light stimuli[23]. Low (70 db) and high (100 db) amplitude acoustic stimuli were delivered using push-style solenoids (12 V) to tap a custom built acrylic stage where the 96-well plate was placed. Acoustic stimuli were recorded using a contact microphone (Aquarian Audio Products, model# H2a) and the freeware audio recording software Audacity (http://www.audacityteam.org). Digital acoustic stimulus was generated as a 70 ms sine wave at various frequencies using Audacity. A computer was used to playback the audio stimulus as an mp3 using an APA150 150 W powered amplifier (Dayton Audio) played through surface transducers adhered to the acrylic stage. Stimulus volume was measured using a BAFX 3608 digital sound level

meter (BAFX Products). Light stimuli were delivered using high intensity LEDs (LEDENGIN) at violet (400 nm, 11 $\mu W/mm^2$), blue (560 nm, 18 $\mu W/mm^2$), and red (650 nm, 11 $\mu W/mm^2$) wavelengths. Stimuli and digital recordings were applied to the entire 96-well plate simultaneously. Instrument control and data acquisition were performed using custom scripts written in MATLAB and Python. The zebrafish motion index (MI) was calculated as follows: MI = sum(abs(frame$_n$ − frame$_{n-1}$)). Normalized MI (nMI) was calculated as follows: nMI = (MI−min(MI))/max(MI). Startle magnitude was calculated using numerical integration via the trapezoidal method (Matlab function *trapz*) of MI values during stimulus.

**Computing the phenoscore**. To quantify distances between multi-dimensional behavioral profiles, we first defined a prototypic behavioral profile to compare everything else against. Etomidate's prototypical behavioral profile was determined from 36 replicates wells treated with etomidate (6.25 $\mu M$) on 3 different plates (12 replicates per plate). Using a simulated annealing procedure (described in the supplement) we identified 12 replicate profiles with the most consistent eASR response that was also most distant from the control (DMSO) wells. The reference profile was the average of these 12 profiles. Phenoscore distances were computed between each well and the reference profile by calculating the correlation distance (using the correlation distance module from the scipy package in python). The correlation distance (phenoscore) has a range from −1 to +1. Positive and negative values represent positive and negative correlation, respectively. Negative values represent anti-correlation. Experimentally, phenoscores tended to saturate at around 0.7, a value that represents substantial positive correlation given that the MI time series is a large vector with >10,000 values. Although etomidate and propofol are both anesthetic GABA$_A$R PAMs with similar behavioral profiles in zebrafish, etomidate is more soluble than propofol, so we used etomidate as the archetypal positive control.

**Ranking the screening hits**. Phenoscores were computed to assign each compound in the screening library a rank order. Hit compounds were defined as the top 125 scoring compounds from this ranked list

**Calculating response magnitude Z-scores**. Response magnitudes were calculated by averaging the maximum motion index value during 3 repeated violet stimuli or 6 repeated acoustic stimuli. These Motion index magnitudes were converted to Z-scores using the following equation: Z-score = (magnitude − mean)/SD. Z-scores were then normalized from 0-1 using the scikit function sklearn.preprocessing.normalize written for python.

**In vitro receptor profiling**. In vitro binding assays and Ki data were generated by the National Institute of Mental Health's Psychoactive Drug Screening Program (PDSP), contract no. HHSN-271-2008-00025-C (NIMH PDSP), for assay details: http://pdsp.med.unc.edu/PDSP%20Protocols%20II%202013-03-28.pdf. Normalized Ki (npKi) values were generated as follows: npKi = 4 + (−log10 (Ki))[64].

**FLIPR**. We used the FLIPR system (Molecular Devices) to quantify GABA-evoked activity of human GABAARs. We chose a membrane potential dye (Molecular Devices) to measure changes in membrane potentials and stably transfected HEK293 cells that expressed $\alpha_1$, $\beta_2$ and $\gamma_2$. Since we observed an increase in GABA-evoked responses when transfected with $\gamma_2$ transiently, we describe the cells as having a low level of γ-subunit expression, indicating heterogeneity of GABA$_A$R compositions in the cell ($\alpha_1\beta_2$ or $\alpha_1\beta_2\gamma_2$). To assay for direct agonists, fluorescence was subtracted pre- and post compound addition. To assay for PAMs, cells were treated with compound at 20 uM and then with 5 uM GABA.

**Whole-brain activity mapping**. Following behavioral experiments, animals were immediately fixed in 4% paraformaldehyde in PBS and incubated overnight at 4 °C. Larvae were then washed in PBS + 0.25% Triton-X (PBT), incubated for 15 min at 70 °C in 150 mM Tris-HCl, pH9, washed in PBT, pearmeablized in 0.05% Trypsin-EDTA for 30-45 min on ice and washed with PBT. Animals were then blocked for 1 h at room temperature (PBT, 1% bovine serum albumin, 2% normal goat serum, 1% DMSO, and 0.02% sodium azide)[38]. The following primary antibodies were diluted into blocking buffer and incubated overnight at 4 °C: α-5HT (1:500, ImmunoStar), α-tERK (1:750, Cell Signaling), α-pERK (1:750, Cell Signaling). Secondary fluorescent antibodies (Life Technologies) were used at 1:500 and incubated in blocking buffer overnight at 4 °C in the dark. Whole-mount fluorescent images were obtained using a Leica SP8 confocal microscope. Image processing was performed in imageJ. Image registration was performed using the Computational Morphometry Toolkit (https://www.nitrc.org/projects/cmtk) and a user interface with the command string defined by Owen Randlett (-awr 010203 -T 8 -X 52 -C 8 -G 80 -R 3 -A '--accuracy 0.4' -W '--accuracy 1.6'). Multiple brains from each condition were then averaged using Matlab scripts to obtain a representative neural activity image. Brightness and contrast were adjusted using Fiji (imageJ). MAP-map calculations (whole-brain ΔpERK significance heat maps) were performed using the analysis code for MAP-map which can be downloaded from the website (http://engertlab.fas.harvard.edu/Z-Brain/).

**Cortisol detection assay**. Cortisol levels were measured in zebrafish[65]. Briefly, 15, 7-day old larvae were treated with the indicated compounds for 1 h. Larvae were anesthetized in ice-cold egg water and then snap-froze in an ethanol/dry ice bath. Larvae were then homogenized in 100 $\mu L$ of water. Cortisol was extracted from the homogenate with 1 mL of ethyl acetate, the resulting supernatant was collected and the solvent allowed to evaporate. Cortisol was dissolved in 0.2% bovine serum albumin (A7030, Sigma) and frozen at −20 °C. For cortisol ELISA experiments, 96-well plates (VWR International) were treated with cortisol antibody (P01-92-94M-P, EastCoast Bio; 1.6 g/mL in PBS) for 16 h at 4 °C, washed, and blocked with 0.1% BSA in PBS. Cortisol samples and cortisol-HRP (P91-92-91H, EastCoast Bio) were incubated at room temperature for 2 h and washed extensively with PBS containing 0.05% Tween-20 (Invitrogen). Detection of HRP was performed using tetra-methylbenzidine (TMB: 22166-1, Biomol) and Tetrabutylammonium borohydride (TBABH: 230170-10 G, Sigma). Reaction was stopped using 1 M $H_2SO_4$. Absorbance was read at 450 nm in an ELISA plate reader (SpectraMax MS, Molecular Devices).

**Software**. Data acquisition and analysis were carried out using custom scripts in Matlab (MathWorks) and Python. Figures were prepared using Matlab, Python, ImageJ (NIH), Prism (GraphPad), and Adobe Illustrator.

**Reporting summary**. Further information on research design is available in the Nature Research Reporting Summary linked to this article.

## Data availability

The source data for all Figures and Supplementary Figs. in the current study are available in the Zenodo repository: https://doi.org/10.5281/zenodo.3336616

## Code availability

Scripts used for data acquisition and analysis are available from the corresponding author upon reasonable request. For Enrichment Factor calculations, code is available upon request.

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

## Acknowledgements

We thank members of our research groups for helpful advice and critical reading of the manuscript. Dr. Owen Randlett for his assistance with the pERK immunostaining and whole-brain imaging protocol. This work was supported by US National Institutes of Health (NIH) grants: R01AA022583 (D.K.), the Paul G. Allen Family Foundation (D.K. and M.K.), R01MH115705 (S.T.), and U01MH104984 (S.T.).

## Author contributions

M.N.M. characterized the phenotype, designed and performed most of the experiments including behavioral assays, compound testing, cortisol detection, imaging, and immunohistochemistry and analyzed the data. L.G. analyzed the data and predicted targets using SEA. R.K. designed, collected, and analyzed the data related to most of the GABAAR reference set. J.T., A.C., and C.R. collected behavioral data. G.B. ran the initial compound screen with support from R.T.P. D.M.T. and C.H. wrote code to collect and

analyze the data. H.J.K. performed the target binding assays and analyzed the data. S.T. performed the FLIPR assays. J.H.G. and J.K.S. synthesized the isoflavones. M.J.K. and D.K. designed the experiments and wrote the paper with input from all the authors. All authors reviewed and edited the manuscript.

## Additional information

**Competing interests:** The authors declare no competing interests.

