## [Peer Review File · Nature Communications]

This manuscript has been previously reviewed at another journal that is not operating a transparent peer review scheme. This document only contains reviewer comments and rebuttal letters for versions considered at Nature Communications.

Reviewers' comments:

Reviewer #1 (Remarks to the Author):

All concerns and requests have now been satisfactorily addressed. The authors should be congratulated on a well-performed and analyzed study that brings to the field a new assay and model system to study the intriguing phenomenon of paradoxical excitation.

Reviewer #4 (Remarks to the Author):

Note that this reviewer chiefly comments on the interpretation of the findings in the context of GABA-A receptor pharmacology.

Comments and Remarks: Generally, a number of clarifications and corrections are needed to correctly depict the putative assignment of some of the hit compounds to members of the GABA-A receptor family and to avoid overstatements and misleading interpretations of the data. Note that "receptor subtype" refers to different compositions and arrangements of 1- 5 subunits into a given receptor pentamer;, while "isoform" refers to isoforms of subunits only. (i.e. beta1, beta2S, beta2L etc. are beta subunit isoforms, whereas alpha1beta2 assemblies and alpha1beta2gamma2 assemblies are receptor subtypes, as per IUPHAR recommendation). Thus, this reviewer uses the term "receptor subtypes" in contrast to the authors who used "receptor isoforms" to mean the same entity. Note also that alpha1beta2 is used in the review text instead of alpha1beta2 etc., as from the context it should be clear what is meant.

Review: The workflow employed by the authors started with the observation that two sedative general anesthetics, that target a wide range of GABA-A receptor subtypes, cause in the zebrafish larvae a phenotype possibly corresponding to paradoxical excitation, namely an enhanced acoustic startle response (eASR) which was not observed for many other "CNS- depressants/ sedatives". In a subsequent testing of a broader panel of GABA-A R targeting compounds (covering orthosteric agonists, allosteric (functional) agonists (also called GABA mimetics) and positive allosteric modulators was done. Apparently allosteric negative modulators were not considered. Among the tested benzodiazepines, midazolam was sadly not tested – in spite of many reports of serious/ severe paradoxical reactions in humans (adolescent and adult patients). Several tested GABA-A receptor ligands displayed also eASRs, while others did not. The definition of the thresholds (dashed lines in panels 1a and 1g) is only qualitatively described in the methods and could be questioned. After having established eASRs as a readout of some, but not all GABA-A receptor targeting compounds (covering anesthetics and compounds with no anesthetic effect, such as valerenic acid), the authors screened >9000 compounds for "etomidate mimicry" assessing immobility and eARS. A largely computational workflow was employed to cluster compounds into structurally similar/ related groups, and representative compounds were re-tested in different downstream protocols: Assessment at cells expressing a mixture of two human GABA-A R subtypes (FLIPR assay), candidate target identification for compounds that tested negative in the FLIPR assay, further downstream testing in some target candidates, further in vivo testing to identify the brain regions in which activation correlated with the eASR phenotype, and some testing for possible anti-nociceptive-like effects. For some compounds that were identified as GABA-A R PAMs at one or both of the tested subtypes, derivatization was performed and further characterization of the derivatives was done. The authors concluded that the eASR assay performed very well to identify compounds with similar pharmacological profiles as etomidate (or propofol) and lacked the etomidate specific adverse effect (lowering cortisol levels). All in all, a massive experimental and computational effort went into the study which has been revised already. While apparently the 2nd revision addressed a number of concerns by other reviewers, this reviewer has been newly recruited and specifically asked to comment on all claims and conclusions concerning GABA-A receptor pharmacology. Indeed, the manuscript as it stands raises many questions, and some major weaknesses need attention before this reviewer sees it suitable for publication. The issues are listed below point by point. After all the discussion of GABA-A

receptor pharmacology is improved to reasonable compliance with IUPHAR terminology recommendations and over-interpretations as well as misleading and premature conclusions are removed from the manuscript, it will provide very valuable data to the community that should lead to genuinely novel insights concerning the phenomenon of "paradoxical excitement" and follow up on the eARS eliciting compounds should lead to much improved insights into its molecular substrates.

Major issues:

1. In the big picture, it seems odd to select a readout that mimics an unwanted drug effect (paradoxical excitation of a sedative) to identify analogues. This apparently has been corrected to some degree by re-focusing the manuscript on the phenomenon of paradoxical excitation as such, and to use the results to better understand eASRs.
2. While in zebrafish the homology of the GABA-A R family with the mammalian family is high, some substantial differences exist and should be mentioned – e.g. eight alpha isoforms in zebrafish versus six in mammals (<https://doi.org/10.1371/journal.pone.0196083>)
3. In Fig 1a, the compounds are grouped into "sedative classes 1-11", the classes are not explained and no reference for the categorization is given. Oddly, we find zolpidem and muscimol in the same class, which is contrary to any pharmacological classification which considers muscimol as an orthosteric agonist and zolpidem a non-benzodiazepine ligand of the benzodiazepine site, and thus a positive allosteric modulator (PAM). Benzodiazepines are given in "class 2". The classification either must be explained and justified, or should be eliminated. In the legend to Figure 1, the term "GABA agonist" is used synonymously with "GABA-A receptor PAM", which is incorrect. The supplement should include a clarification of terminology, and current IUPHAR recommendations should be followed consistently (with historical terms such as benzodiazepine receptor agonist in parenthesis where they may be appropriate).
4. Legend for Fig 1g: The class descriptions here are also somewhat murky, the authors claim that group 6 contains "other PAMs and direct agonists". A category "other compounds with mixed or unspecified functional activity" would be preferred – several of the other classes also contain "GABA-mimetics" (functional agonists).
5. Figure 2g, interpretation of the phenoscore: As mentioned above, the phenoscore is described qualitatively in the methods, a more rigorous explanation would be helpful. Moreover, it is based on referencing to etomidate's efficacy to elicit eASRs. The conclusion that benzodiazepines generally do not elicit eASRs seems not completely valid for two reasons: Reason 1, benzodiazepines noted to cause paradoxical reactions in humans (chiefly midazolam) were not tested; and Reason 2, diazepam elicited some responses directly at the threshold which divides the y-axis into eASR and non- eASR. I suggest to categorize into at least three categories, as for quite many compounds a very broad range with both positive and negative responses are seen, while other compounds elicit much more consistent patterns. (Perhaps the phenomenon of paradoxical excitatory responses in the zebrafish (larvae) is also inconsistent across individuals, as it is in humans?) Some statistical analysis would provide a deeper insight compared to the binary classification.
6. Text, p. 5 "the data suggest that a non-gamma isoform is involved" ... "activation of both γ -containing and δ -containing isoforms contributes..." This passage is still extremely misleading and not supported by the data at all. The only conclusion that can legitimately be drawn from the results displayed in Figure 2g is that the known in vitro pharmacologies of the compounds that elicit eASRs is widely diverse and does not clearly point to any subset of receptor subtypes that mediates the effect. The effect seems limited to allosteric ligands - but the number of orthosteric ligands that was tested is likely too small for final conclusions (likewise, the number of benzodiazepines).
7. Study design/ flaw in study design: If the authors indeed assume that "a non-gamma isoform is involved" in eliciting eASRs, the FLIPR assay with a cell line expressing $\alpha 1$, $\beta 2$ and (to a lower degree) $\gamma 2$ subunits seems the wrong assay. While it is very interesting that a high percentage of the hits that produce eASRs are PAMs of $\alpha 1\beta 2$ or $\alpha 1\beta 2\gamma 2$ GABA-A receptors, the finding per se is not helpful for the identification of receptor subtypes that mediate the eASRs.
8. FLIPR assay control compounds, text on p6: "activation" and "positive allosteric modulation" are

used interchangeably, but are different phenomena. See also point 3, the authors should define their favorite terminology and use it consistently. More importantly, the "negative control compounds" are not consistently inactive or silent modulators, instead, the channel blocker picrotoxin has also been used. This seems an odd choice. Moreover, some test compounds seem to have displayed negative allosteric modulation (NAM), this should be discussed.

9. P6, "non-GABAAR targets": The authors state that "hit compounds that did not act on GABAARs ..." – this should be reworded to "did not show PAM or agonist activity in the FLIPR assay at the $\alpha 1\beta 2/ \alpha 1\beta 2\gamma 2$ receptor mixture" ... "which does not exclude effects at other receptor subtypes". As mentioned above, if the authors suspect that non- $\gamma 2$ containing receptor subtypes contribute to eARS, it is very odd to test the "canonical" diazepam sensitive receptor in the in vitro follow up. The limitations that come from testing only in this one assay should be discussed.

10. Discussion (blue part of the discussion): As indicated above in point 6, I disagree with the authors on the conclusion concerning "non- $\gamma 2$ isoforms" being the most likely candidates for the mediation of the eASR phenotype. All the compounds that were used here are highly polyspecific (i.e. modulate a large number of receptor subtypes), and their in vitro subtype profiles are incomplete owing to the large number of receptor subtypes (by far exceeding those often tested in recombinant systems). A different view could also be that the efficacy in different beta isoforms is what drives the presence or absence of eASRs (see e.g. doi: 10.1186/1471-2210-7-2 and doi: 10.1124/jpet.109.161885). Just as easily possible, perhaps PAM effects at some receptor subtypes have a "dominant" or protective effect, while PAM (or even NAM) effects at other subtypes trigger the eASRs if a compound is inactive or less active at the "dominant immobilizing" subtypes. Thus, the relative efficacies at different subtypes may lead to dominant immobilization or dominant startle behavior.

Remark: The distinction between PAM (activating) and inactive or NAM may even fall short of physiological relevance, as PMAs can either leave desensitization kinetics unchanged, or accelerate desensitization (in which case they would only transiently increase charge transfer, but in sum actually reduce net charge transfer and thus not enhance but diminish GABA effects). Given all this complexity, I feel that the authors should NOT suggest any receptor pool, but instead suggest that the effects of the eASR eliciting compounds should be studied in a much wider panel of defined receptor subtypes, and that the subunits present in the activated hindbrain neurons should be identified to shed more light on the issue.

11. Discussion, "preferential activity at delta-containing GABA-A receptors": It has been a widely used procedure to compare enhancement of a reference current across different receptor subtypes. A recent paper raises serious concerns about the validity when comparing subtypes with large average current levels with those that display "partial agonistic responses to GABA" or much lower GABA- elicited average current levels: doi: 10.1016/j.phrs.2016.05.014. To provide a balanced view of "delta preference" versus the view that modulation should be assessed at comparable opening probabilities, in which case the "delta preference" is no longer seen, this work should be cited and the controversy mentioned.

12. P 9, the headline advertises "improvement" of hit compounds. I suggest to delete "improvement" or to replace with "derivatization of selected hit compounds". Since it is totally unclear which properties will be improved in addition to the absence of effect on cortisol levels, this chapter seems to represent valuable but early pilot work towards an SAR insight. The results obtained in this would should also be discussed in the discussion section: While the varying in vivo activity of the 21 analogs of compound 7013338 points towards very steep SAR cliffs governing activity on a key target species, this key target (one or several GABA-A receptor subtypes in all likelihood) would need to be identified by testing activity with a more sensitive (e.g. electrophysiological) functional assay of recombinantly expressed receptor subtypes. Such prospects deserve to be discussed as an outlook.

Minor points:

In a number of places the language is not clear and multiple interpretations of the text are possible:

- P5, "we tested a variety of ligands at different receptor subtypes" – NO testing at different

receptor subtypes was done, please reword to clarify what is meant.

- P 6 "... identify compounds with conserved activity against human GABAARs" - what is "conserved activity"?? Should it read "suggesting conserved GABA-A R pharmacology between zebrafish and mammals"?

Rebuttal letter

Manuscript Title: Zebrafish behavioral profiling identifies ligands, targets, and neurons related to sedation and paradoxical excitation

Dear Editor and Reviewers,

Thank you for reviewing our manuscript. The reviews raised many questions, comments, and suggestions and we appreciate the opportunity to address them. While the reviews were mostly positive, there were at least two major issues.

One major issue, raised by Reviewer 4, was that we overinterpreted the GABA_AR ligand data. Reviewer 4 raised several insightful and alternative interpretations. We agree with the Reviewer's comments and have revised our original claims about benzodiazepines and gamma-containing receptors. Whereas the original manuscript claimed that benzodiazepines did not phenocopy etomidate, the revised manuscript shows that benzodiazepines cause weak and intermediate phenotypes. To support these claims, we have included new statistical analyses (**Fig. S20**), new thresholds (**Fig. 1g**), and new motor activity plots (**Fig. S2**). In addition, we have added substantial new text discussing the phenotypes caused by benzodiazepines, and elaborating on the strengths and limitations of the pharmacology data (**as described in Issue 5**).

Another major issue, was the need to clarify inconsistent and potentially misleading terminology. For example, the original manuscript conflated the terms 'subtype' and 'isoform' and did not define the GABA_AR naming conventions used in the paper. In the revised manuscript, the terminology has been updated to conform with IUPHAR naming standards as described in the **Supplemental Methods** (see reply to **Issue 3**).

In addition, the revised manuscript contains further revisions responding to all Reviewers' comments and concerns, as described point by point below.

Reviewers' comments:

Reviewer #1 (Remarks to the Author):

All concerns and requests have now been satisfactorily addressed. The authors should be congratulated on a well-performed and analyzed study that brings to the field a new assay and model system to study the intriguing phenomenon of paradoxical excitation.

Reply: We thank the Reviewer for their thoughtful and constructive feedback throughout the review process.

Reviewer #4 (Remarks to the Author):

Note that this reviewer chiefly comments on the interpretation of the findings in the context of GABA-A receptor pharmacology.

Comments and Remarks:

Generally, a number of clarifications and corrections are needed to correctly depict the putative assignment of some of the hit compounds to members of the GABA-A receptor family and to avoid overstatements and misleading interpretations of the data. Note that “receptor subtype” refers to different compositions and arrangements of 1- 5 subunits into a given receptor pentamer, while “isoform” refers to isoforms of subunits only. (i.e. beta1, beta2S, beta2L etc. are beta subunit isoforms, whereas $\alpha 1\beta 2$ assemblies and $\alpha 1\beta 2\gamma 2$ assemblies are receptor subtypes, as per IUPHAR recommendation). Thus, this reviewer uses the term “receptor subtypes” in contrast to the authors who used “receptor isoforms” to mean the same entity. Note also that $\alpha 1\beta 2$ is used in the review text instead of $\alpha 1\beta 2$ etc., as from the context it should be clear what is meant.

Reply: We thank the Reviewer for providing specific feedback about how to improve these issues. We appreciate their time and advice and have revised the manuscript point-by-point below.

Review: The workflow employed by the authors started with the observation that two sedative general anesthetics, that target a wide range of GABA-A receptor subtypes, cause in the zebrafish larvae a phenotype possibly corresponding to paradoxical excitation, namely an enhanced acoustic startle response (eASR) which was not observed for many other “CNS- depressants/ sedatives”. In a subsequent testing of a broader panel of GABA-A R targeting compounds (covering orthosteric agonists, allosteric (functional) agonists (also called GABA mimetics) and positive allosteric modulators was done. Apparently allosteric negative modulators were not considered. Among the tested benzodiazepines, midazolam was sadly not tested – in spite of many reports of serious/ severe paradoxical reactions in humans (adolescent and adult patients). Several tested GABA-A receptor ligands displayed also eASRs, while others did not. The definition of the thresholds (dashed lines in panels 1a and 1g) is only qualitatively described in the methods and could be questioned. After having established eASRs as a readout of some, but not all GABA-A receptor targeting compounds (covering anesthetics and compounds with no anesthetic effect, such as valerenic acid), the authors screened >9000 compounds for “etomidate mimicry” assessing immobility and eARS. A largely computational workflow was employed to cluster compounds into structurally similar/ related groups, and representative compounds were re-tested in different downstream protocols: Assessment at cells expressing a mixture of two human GABA-A R subtypes (FLIPR assay), candidate target identification for compounds that tested negative in the FLIPR assay, further downstream testing in some target candidates, further in vivo testing to identify the brain regions in which activation correlated with the eASR phenotype, and some testing for possible anti-nociceptive-like effects. For some compounds that were identified as GABA-AR PAMs at one or both of the tested subtypes, derivatization was performed and further characterization of the derivatives was done. The authors concluded that the eASR assay performed very well to identify compounds with similar pharmacological profiles as etomidate (or propofol) and lacked the etomidate specific adverse effect (lowering cortisol levels). All in all, a massive experimental and computational effort went into the

study which has been revised already. While apparently the 2nd revision addressed a number of concerns by other reviewers, this reviewer has been newly recruited and specifically asked to comment on all claims and conclusions concerning GABA-A receptor pharmacology. Indeed, the manuscript as it stands raises many questions, and some major weaknesses need attention before this reviewer sees it suitable for publication. The issues are listed below point by point. After all the discussion of GABA-A receptor pharmacology is improved to reasonable compliance with IUPHAR terminology recommendations and over-interpretations as well as misleading and premature conclusions are removed from the manuscript, it will provide very valuable data to the community that should lead to genuinely novel insights concerning the phenomenon of “paradoxical excitement” and follow up on the eARS eliciting compounds should lead to much improved insights into its molecular substrates.

Major issues:

1. In the big picture, it seems odd to select a readout that mimics an unwanted drug effect (paradoxical excitation of a sedative) to identify analogues. This apparently has been corrected to some degree by re-focusing the manuscript on the phenomenon of paradoxical excitation as such, and to use the results to better understand eASRs.

Reply 1. We appreciate the Reviewer’s feedback and have updated the abstract to focus on paradoxical excitation.

The revised abstract states:

Anesthetics are generally associated with sedation, but some anesthetics can also increase brain and motor activity — a phenomenon known as paradoxical excitation. Previous studies have identified GABA_A receptors as the primary targets of most anesthetic drugs, but how these compounds produce paradoxical excitation is poorly understood.

2. While in zebrafish the homology of the GABA-A R family with the mammalian family is high, some substantial differences exist and should be mentioned – e.g. eight alpha isoforms in zebrafish versus six in mammals (<https://doi.org/10.1371/journal.pone.0196083>)

Reply 2. We agree with the Reviewer that it is important to emphasize differences between the species.

The revised text states (**Introduction, pg. 4**):

Additionally, there are several important differences between the species. One such difference is that the zebrafish genome encodes a GABA_AR β₄-subunit, which does not have a clear

ortholog in mammals¹⁹. Another difference is that whereas mammals have six GABA_AR α -subunit isoforms, zebrafish have eight²⁰.

3. In Fig 1a, the compounds are grouped into “sedative classes 1-11”, the classes are not explained and no reference for the categorization is given. Oddly, we find zolpidem and muscimol in the same class, which is contrary to any pharmacological classification which considers muscimol as an orthosteric agonist and zolpidem a non-benzodiazepine ligand of the benzodiazepine site, and thus a positive allosteric modulator (PAM). Benzodiazepines are given in “class 2”. The classification either must be explained and justified, or should be eliminated. In the legend to Figure 1, the term “GABA agonist” is used synonymously with “GABA-A receptor PAM”, which is incorrect. The supplement should include a clarification of terminology, and current IUPHAR recommendations should be followed consistently (with historical terms such as benzodiazepine receptor agonist in parenthesis where they may be appropriate).

Reply 3. We agree with the Reviewer. We moved muscimol from panel a to panel g, and we added citations to **Supplemental Table 1**. In addition, we revised the terminology to clearly distinguish between agonists and PAMs in the figure legend.

Additional text was added to clarify the GABA_AR terminology (**Supplemental Methods, pg.3**):

GABA_AR terminology. Notation for GABA receptors conform to IUPHAR recommendations (Barnard et al. 1998). Receptor subunits are indicated by their greek symbols with subscripted numbers to indicated specific isoforms as in: “the α_1 subunit isoform”. To refer to GABA_A receptor (GABA_AR) subtypes, the term GABA is used to indicate the receptor type, and the subscript A is used to refer to all GABA_ARs. Subtypes comprised of specific subunit isoforms are indicated like: “the $\alpha_1\beta_2\gamma_2$ subtype.”

4. Legend for Fig 1g: The class descriptions here are also somewhat murky, the authors claim that group 6 contains “other PAMs and direct agonists”. A category “other compounds with mixed or unspecified functional activity” would be preferred – several of the other classes also contain “GABA-mimetics” (functional agonists).

Reply 4. We agree with the Reviewer. The problematic classification for category 6 has been eliminated (**Fig 1g**). Revised classifications (with citations) are now listed in the new **Table S3**.

5. Figure 2g, interpretation of the phenoscore: As mentioned above, the phenoscore is described qualitatively in the methods, a more rigorous explanation would be helpful. Moreover, it is based on

referencing to etomidate's efficacy to elicit eASRs. The conclusion that benzodiazepines generally do not elicit eASRs seems not completely valid for two reasons: Reason 1, benzodiazepines noted to cause paradoxical reactions in humans (chiefly midazolam) were not tested; and Reason 2, diazepam elicited some responses directly at the threshold which divides the y-axis into eASR and non- eASR. I suggest to categorize into at least three categories, as for quite many compounds a very broad range with both positive and negative responses are seen, while other compounds elicit much more consistent patterns. (Perhaps the phenomenon of paradoxical excitatory responses in the zebrafish (larvae) is also inconsistent across individuals, as it is in humans?) Some statistical analysis would provide a deeper insight compared to the binary classification.

Reply 5. We agree with the Reviewer. Although the phenoscore facilitates comparisons of multidimensional phenotypes, we should have done a more rigorous statistical analysis on the results. Rather than applying a simple binary classification and claiming that the benzodiazepine data do not phenocopying etomidate, we reanalyzed the data with thresholds derived from a new statistical analysis described in Supplementary Methods. The Reviewer was correct to notice that diazepam was right on the border of the previous threshold, and these analyses support the Reviewer's insight that benzodiazepines cause interesting intermediate phenotypes.

To report these phenotypes, we added new thresholds (**Fig. 1g**), new plots showing example phenotypes caused by diazepam (**Fig. S2**), and *P* values for the comparisons. For example, although by the phenoscore metric the diazepam profile is significantly unlike etomidate ($P < 1e^{-6}$) the example plots illustrate that diazepam does produce an interesting intermediate phenotype.

The revised manuscript also contains substantial new text including 1) a more accurate description of the benzodiazepine results with statistical support 2) a revised discussion explaining why these data do not rule out gamma-containing subtypes 3) a rigorous explanation of the phenoscore, threshold determination, and significance.

1. Results, pg. 5

To determine if other GABA_AR ligands caused similar phenotypes, we used the phenoscore metric to quantify similarities between the archetypal profile caused by etomidate (6.5 μ M) and a diverse range of GABAergic compounds (**Supplementary Table 3**). Average phenoscores of DMSO-treated negative controls were significantly less than etomidate-treated positive controls (0.2 versus 0.71, $P < 10^{-10}$) (**Fig. S20**). For the test compounds, average phenoscores fell on a continuum between the positive and negative controls (**Fig. 1g**). Based on statistical simulations, we subdivided these phenoscores into three categories: weak, intermediate, and strong.

Compounds with weak phenoscores ($x < 0.51$) included one GABA_B receptor agonist, one PAM of δ -subunit containing GABA_ARs, two non-BZ-site ligands, three structurally-related GABA_AR orthosteric agonists, and seven BZ-site GABA_AR PAMs (**Fig. 1g**). For these

compounds, the average phenoscores were significantly less than the positive controls ($P < 0.01$, **Fig. S20a**), suggesting that these compounds did not phenocopy etomidate. For example, the highest scoring ocinaplon treatment produced a behavioral profile that resembled the negative controls (**Fig. S2**). These data suggest that a variety of GABAergic compounds do not cause sedation and paradoxical excitation.

Compounds with intermediate phenoscores ($0.51 < x < 0.71$) included several types of GABA_AR PAMs including thiopental, carboetomidate, THDOC, alfaxalone, diazepam, and valerenic acid. The highest scoring profiles produced by some of these compounds (including alfaxalone, thiopental, and tracazolote) showed a barely detectable statistically significant difference compared to the positive controls ($0.01 < P < 0.05$). The highest scoring profiles of animals treated with diazepam, and valerenic acid were significantly lower than the positive controls ($P < 0.01$, **Fig. S20a**), however these treatments produced interesting intermediate effects on sedation and paradoxical excitation. For example, although the highest-scoring diazepam treatment was strongly sedating in most assays, it produced eASRs that were relatively weak and inconsistent (**Fig. S2**). These data suggest that a variety of PAMs have intermediate effects on sedation and paradoxical excitation.

Finally, compounds with strong phenoscores ($0.71 < x < 1$) included several anesthetic and neurosteroid PAMs including etomidate, propofol, progesterone, and DOC (**Fig. 1g**). The highest scoring treatments for these compounds produced behavioral profiles that were both strongly sedating and produced high-magnitude eASRs (**Fig. S2**). These profiles were not statistically different from the positive controls ($P > 0.05$, **Fig. S20**). Interestingly, although DOC and progesterone are neurosteroid precursors, they were among the most potent compounds tested (**Fig. 1g**). As expected, progesterone's etomidate-like phenotype was suppressed by dutasteride, a 5- α -reductase inhibitor that blocks the metabolic conversion of progesterone to allopregnanolone, suggesting that these compounds were converted to active neurosteroids (**Fig. S5**). Together, these data suggest that a subset of GABA_AR PAMs cause sedation and paradoxical excitation in zebrafish. However, due to the overlapping pharmacology of numerous GABA_AR subtypes, these data do not clearly point to any specific subset of receptor subtypes as being necessary or sufficient for these behaviors.

2. Discussion, pg. 13

Although these studies show that GABA_AR PAMs cause paradoxical excitation, pharmacological experiments to determine which GABA_AR subtypes caused eASRs were ultimately inconclusive. While the majority of GABA_ARs in the CNS are benzodiazepine-sensitive γ -containing subtypes, and multiple benzodiazepines did not cause strong eASRs (**Fig. 1g**), γ -containing subtypes may still be very important for eASRs. One reason is that the benzodiazepines tested in this study only represent a very small subset of benzodiazepine analogs. Another reason is that diazepam produced intermediate eASR phenotypes (**Fig. 1g, S2**), suggesting that other benzodiazepines may cause even stronger

eASR phenotypes. Although etomidate, propofol, neurosteroids, and other anesthetics are PAMs at δ -subunit containing GABA_AR subtypes, these ligands also modulate γ -containing subtypes. Furthermore, although THIP and DS2 are reported to have preferential activity at δ -containing GABA_ARs, these compounds also modulate γ -containing receptors²⁹ and did not cause eASRs. One alternative explanation is that β -isoforms^{62,63} could drive the presence or absence of eASRs. Another possible explanation is that whereas PAMs may produce immobilizing effects via some receptor subtypes, they may produce eASRs via other subtypes. In summary, although a subset of GABA_AR PAMs caused eASRs, these compounds may do so via a variety of receptor subtypes. In future studies, it would be interesting to test additional benzodiazepines for such effects including midazolam which causes paradoxical excitation in humans⁶⁴. The specificity of currently available pharmacological tools may be insufficient to determine which GABA_AR subtypes cause eASRs. Therefore, future studies may require targeted knockouts and other genetic tools to help identify the key receptor subtypes.

3. **Methods, pg. 15.**

Computing the phenoscore. To quantify distances between multi-dimensional behavioral profiles, we first defined a prototypic behavioral profile to compare everything else against. Etomidate's prototypical behavioral profile was determined from 36 replicates wells treated with etomidate (6.25 μ M) on 3 different plates (12 replicates per plate). Using a simulated annealing procedure (described in the supplement) we identified 12 replicate profiles with the most consistent eASR response that was also most distant from the DMSO control wells. The reference profile was the average of these 12 profiles. Phenoscore distances were computed between each well and the reference profile by calculating the correlation distance (using the correlation distance module from the scipy package in python). The correlation distance (phenoscore) has a range from -1 to +1. Positive and negative values represent positive and negative correlation, respectively. Negative values represent anti-correlation. Experimentally, phenoscores tended to saturate at around 0.7, a value that represents substantial positive correlation given that the MI time series is a large vector with >10,000 values. Although etomidate and propofol are both anesthetic GABA_AR PAMs with similar behavioral profiles in zebrafish, etomidate is more soluble than propofol, so we used etomidate as the archetypal positive control.

4. **Supplemental Methods, pg 2**

Determination of phenotypic thresholds and significance. For each ligand, we selected the dose that gave the highest average phenoscore, and for that dose, we performed a two-sample Kolmogorov-Smirnov (KS) test to calculate the KS statistic against the 12 positive control replicates of etomidate @ 6.25 μ M using the scipy function `ks_2samp` from the `scipy.stats` package (**Fig. S20a**). To calculate approximate thresholds of phenoscore significance, we performed a statistical simulation. For each score in the space of possible phenoscores (binned in 0.05 increments from 0 to 1), we sampled 12 replicates from a uniform distribution centered around the score ranging from -4σ to $+4\sigma$ away from the mean, and

calculated the KS statistic against the etomidate 6.25 μ M replicates. We repeated this simulated procedure 100 times to get robust statistics, and took the average of these P values. However, we realized that the standard deviation of replicates across different GABA_AR ligands was not a constant value. It tended to be low for extremely poor phenotypes, peaked for intermediate phenotypes, and decreased again for extremely strong phenotypes. Therefore, we fit the standard deviations for GABA_AR ligands as a function of phenoscore with a 10th order polynomial using the Polynomial package in numpy (**Fig. S20b**). Using this resulting polynomial, we calculated the KS P values from the simulated uniform distributions as we iteratively stepped along the y-axis ; these P values were smoothly distributed except for a discontinuity around phenoscore 0.5 due to rapidly increasing P values in this range (**Fig. S20c**). We derived the threshold phenoscores associated with these P values by fitting another polynomial to the resulting distribution in the smooth region (above phenoscore 0.5) (**Fig. S20d**) and calculating the roots of the function at those P values. The resulting phenoscores corresponding to 0.01 and 0.05 P value thresholds were 0.51 and 0.71, respectively.

6. Text, p. 5 “the data suggest that a non-gamma isoform is involved” ... “activation of both γ -containing and δ -containing isoforms contributes...” This passage is still extremely misleading and not supported by the data at all. The only conclusion that can legitimately be drawn from the results displayed in Figure 2g is that the known in vitro pharmacologies of the compounds that elicit eASRs is widely diverse and does not clearly point to any subset of receptor subtypes that mediates the effect. The effect seems limited to allosteric ligands - but the number of orthosteric ligands that was tested is likely too small for final conclusions (likewise, the number of benzodiazepines).

Reply 6. We agree with the Reviewer. The claims regarding non-gamma isoforms were revised, as described in **Issue 5**.

7. Study design/ flaw in study design: If the authors indeed assume that “a non-gamma isoform is involved” in eliciting eASRs, the FLIPR assay with a cell line expressing $\alpha 1$, $\beta 2$ and (to a lower degree) $\gamma 2$ subunits seems the wrong assay. While it is very interesting that a high percentage of the hits that produce eASRs are PAMs of $\alpha 1\beta 2$ or $\alpha 1\beta 2\gamma 2$ GABA-A receptors, the finding per se is not helpful for the identification of receptor subtypes that mediate the eASRs.

Reply 7. We agree with the Reviewer. The $\alpha 1\beta 2\gamma 2$ -expressing cell line helps to explain the targets of some novel hit compounds. However, the cell line does not help to identify which subtypes mediate eASRs. Therefore, we eliminated the original claim about the non-gamma isoform, as described in our reply to **Issue 5**.

8. FLIPR assay control compounds, text on p6: “activation” and “positive allosteric modulation” are used interchangeably, but are different phenomena. See also point 3, the authors should define their favorite terminology and use it consistently. More importantly, the “negative control compounds” are not consistently inactives or silent modulators, instead, the channel blocker picrotoxin has also been used. This seems an odd choice. Moreover, some test compounds seem to have displayed negative allosteric modulation (NAM), this should be discussed.

Reply 8. We agree with the Reviewer. Inappropriate use of the term “activation” has now been eliminated. In addition, the revised text includes new text describing the negative controls and the negative change in fluorescence (**Results, pg. 7**):

In this cell line, etomidate, tracazolate, and propofol increased fluorescence in the presence of GABA, as expected for GABA_AR PAMs. In addition, half of the tested hit compounds (23/46) also showed PAM activity (**Fig. 2f; S9**). By contrast, PAM activity was not observed with negative control compounds including BGC 20-761 (an HTR6 antagonist) and PTX (a GABA_AR channel blocker) which likely reduced GABA_AR activity due to inhibition of constitutively active GABA_ARs in the system. Interestingly, the PAM activity of two hit compounds, 7013338 and 5942595, was significantly greater than the positive controls (**Fig. 2f, p < 0.0001, n = 4**). While some of the novel compounds appeared to function in this assay as negative allosteric modulators (NAMs), reductions in fluorescence were likely due to toxicity-induced cell loss (**Fig. 2f, Supplementary Table 5**).

9. P6, “non-GABAAR targets”: The authors state that “hit compounds that did not act on GABAARs ...” – this should be reworded to “did not show PAM or agonist activity in the FLIPR assay at the $\alpha 1\beta 2 / \alpha 1\beta 2\gamma 2$ receptor mixture” ... “which does not exclude effects at other receptor subtypes”. As mentioned above, if the authors suspect that non- $\gamma 2$ containing receptor subtypes contribute to eARS, it is very odd to test the “canonical” diazepam sensitive receptor in the in vitro follow up. The limitations that come from testing only in this one assay should be discussed.

Reply 9. We agree with the Reviewer. The phrase “hit compounds that did not act on GABAARs ...” was eliminated and the claims about receptor subtype selectivity were revised, as described in **Issue 5**. In addition, new we added new text discussing the limitations of the in vitro assay. The revised text states (**Discussion Pg. 12**):

Indeed, these studies may have underestimated the number of hit compounds that targeted GABA_ARs for several reasons. One reason is that the *in vitro* GABA_AR FLIPR assay only tested a very small number of receptor subtypes and subunit isoforms ($\alpha 1\beta 2$ and $\alpha 1\beta 2\gamma 2$). As a result, these assays would have missed compounds that acted on other GABA_AR subtypes. A

second reason is that some of the hit compounds may act on zebrafish-specific GABA_ARs. Finally, some hit compounds that caused eASRs in zebrafish may need to be bioactivated *in vivo*, and would therefore not be active *in vitro*. Therefore, even more of the hit compounds may have targeted GABA_ARs.

10. Discussion (blue part of the discussion): As indicated above in point 6, I disagree with the authors on the conclusion concerning “non-gamma isoforms” being the most likely candidates for the mediation of the eASR phenotype. All the compounds that were used here are highly polyspecific (i.e. modulate a large number of receptor subtypes), and their *in vitro* subtype profiles are incomplete owing to the large number of receptor subtypes (by far exceeding those often tested in recombinant systems). A different view could also be that the efficacy in different beta isoforms is what drives the presence or absence of eASRs (see e.g. doi: 10.1186/1471-2210-7-2 and doi: 10.1124/jpet.109.161885). Just as easily possible, perhaps PAM effects at some receptor subtypes have a “dominant” or protective effect, while PAM (or even NAM) effects at other subtypes trigger the eASRs if a compound is inactive or less active at the “dominant immobilizing” subtypes. Thus, the relative efficacies at different subtypes may lead to dominant immobilization or dominant startle behavior.

Reply 10. We agree with the reviewer. The claims about gamma-containing GABAAR subtypes were revised, as described in our reply to **Issue 5**.

Remark: The distinction between PAM (activating) and inactive or NAM may even fall short of physiological relevance, as PMAs can either leave desensitization kinetics unchanged, or accelerate desensitization (in which case they would only transiently increase charge transfer, but in sum actually reduce net charge transfer and thus not enhance but diminish GABA effects). Given all this complexity, I feel that the authors should NOT suggest any receptor pool, but instead suggest that the effects of the eASR eliciting compounds should be studied in a much wider panel of defined receptor subtypes, and that the subunits present in the activated hindbrain neurons should be identified to shed more light on the issue.

Reply. We agree with the Reviewer. The revised text does not suggest any receptor pool, as described in **issue 5**.

We agree that the compounds should be studied on a wider panel of subtypes, and we are currently looking for to collaborate on the *in vitro* profiling assays. We also agree that the subunits in the activated hindbrain should be identified, and we are genetically targeting candidate subunits in the zebrafish to pursue these lines of inquiry in future studies.

11. Discussion, “preferential activity at delta-containing GABA-A receptors”: It has been a widely used procedure to compare enhancement of a reference current across different receptor subtypes. A recent paper raises serious concerns about the validity when comparing subtypes with large average current levels with those that display “partial agonistic responses to GABA” or much lower GABA-elicited average current levels: doi: 10.1016/j.phrs.2016.05.014. To provide a balanced view of “delta preference” versus the view that modulation should be assessed at comparable opening probabilities, in which case the “delta preference” is no longer seen, this work should be cited and the controversy mentioned.

Reply 11. We agree with the reviewer. The controversy about subtype selectivity is now discussed with the indicated citation, as described in **Issue 5**.

12. P 9, the headline advertises “improvement” of hit compounds. I suggest to delete “improvement” or to replace with “derivatization of selected hit compounds”. Since it is totally unclear which properties will be improved in addition to the absence of effect on cortisol levels, this chapter seems to represent valuable but early pilot work towards an SAR insight. The results obtained in this would should also be discussed in the discussion section: While the varying in vivo activity of the 21 analogs of compound 7013338 points towards very steep SAR cliffs governing activity on a key target species, this key target (one or several GABA-A receptor subtypes in all likelihood) would need to be identified by testing activity with a more sensitive (e.g electrophysiological) functional assay of recombinantly expressed receptor subtypes. Such prospects deserve to be discussed as an outlook.

Reply 12. We agree with the Reviewer. The term “improvement” has been deleted and new text was added to the Discussion to elaborate on the implications of the SAR data. The revised discussion states (**Discussion, p. 14**)

Even if the hit compounds translated to mammals (causing both sedation and paradoxical excitation), additional studies would be necessary to determine if the paradoxical excitation phenotype could be overcome at higher concentrations or via medicinal chemistry. For example, the 21 analogs of compound 7013338 showed variable efficacies (**Fig. 6e**), suggesting that it may be possible to use medicinal chemistry to increase or decrease eASR activity. These shifts likely correspond specific effects on one or more molecular targets that would need to be identified with more sensitive functional assays, such as electrophysiological experiments, of recombinantly expressed receptor subtypes. Future studies may seek to identify different ligands that sedate zebrafish without causing eASRs, or eASRs may be used as a counter screen for other potential anesthetic lead compounds. Presumably, such compounds would work through different mechanisms than etomidate, propofol, and the other

compounds identified in this study, and would further improve our understanding of GABAergic signaling, anesthesia, and paradoxical excitation.

Minor points:

In a number of places the language is not clear and multiple interpretations of the text are possible:

- P5, “we tested a variety of ligands at different receptor subtypes” – NO testing at different receptor subtypes was done, please reword to clarify what is meant.

Reply. We agree with the Reviewer. The unclear phrase was deleted and replaced. The revised text states (**Results, pg. 5**)

To determine if other GABA_AR ligands caused similar phenotypes, we profiled a variety of ligands at a range of concentrations (**Supplementary Table 3**).

- P 6 “... identify compounds with conserved activity against human GABAARs” - what is “conserved activity”?? Should it read “suggesting conserved GABA-A R pharmacology between zebrafish and mammals”?

Reply. We agree with the Reviewer. The term “conserved activity” has been deleted. The revised text states (**Results, pg. 7**):

These data suggest that behavioral screens in zebrafish can enrich for compounds with activity at specific human receptors. In addition, these data suggest that many of the hit compounds identified in the screen cause sedation and paradoxical excitation via GABA_ARs.

REVIEWERS' COMMENTS:

Reviewer #4 (Remarks to the Author):

The authors have done a very comprehensive and convincing job at addressing all concerns expressed by this reviewer.

NCOMMS-19-06703B

Re: Large-scale behavior-based chemical screening identifies ligands and targets related to paradoxical excitation in zebrafish

Dear Editor,

We have fully responded to the Reviewer.

REVIEWERS' COMMENTS:

Reviewer #4 (Remarks to the Author):

The authors have done a very comprehensive and convincing job at addressing all concerns expressed by this reviewer.